# EVERY STEP COUNTS: DECODING TRAJECTORIES AS AUTHORSHIP FINGERPRINTS OF DLLMS

## ABSTRACT

Discrete Diffusion Large Language Models (dLLMs) have recently emerged as a competitive paradigm for non-autoregressive language modeling. Their distinctive decoding mechanism enables faster inference speed and strong performance in code generation and mathematical tasks. In this work, we show that the decoding mechanism of dLLMs not only enhances model utility but also can be used as a powerful tool for model attribution. Specifically, by analyzing the decoding trajectory associated with a given response, we can effectively identify its source model, thereby helping to mitigate risks of harmful content caused by model misuse. A key challenge in this problem lies in the diversity of attribution scenarios, including distinguishing between different models as well as between different checkpoints or backups of the same model. To ensure broad applicability, we identify two fundamental problems: *what information to extract from the decoding trajectory, and how to utilize it effectively.* We first observe that relying directly on per-step model confidence yields poor performance. This is mainly due to the bidirectional decoding nature of dLLMs: each newly decoded token influences the confidence of other decoded tokens, making model confidence highly redundant and washing out structural signal regarding decoding order or dependencies. To overcome this, we propose a novel information extraction scheme called the *Directed Decoding Map (DDM)*, which captures structural relationships between decoding steps and better reveals model-specific behaviors. Furthermore, to make full use of the extracted structural information during attribution, we propose *Gaussian-Trajectory Attribution (GTA)*, where we fit a cell-wise Gaussian distribution at each decoding position for each target model, and define the log-likelihood difference of a trajectory under different distributions as the attribution score: if a trajectory exhibits higher log-likelihood under the distribution of a specific model, it is more likely to have been generated by that model. Extensive experiments under different settings validate the utility of our methods.

## 1 INTRODUCTION

Authorship attribution has long been an important problem in natural language processing, with wide applications in criminal investigations (Chaski, 2005; Koppel et al., 2008; Grant, 2020), tracking terrorist threat (Cafiero & Camps, 2023; Budryk, 2019), and social media protection (Hazell, 2023; Barbon Jr et al., 2017; Sinnott & Wang, 2021). In the era of large language models (LLMs), the value of attribution becomes even more prominent, as it promotes responsibility assignment and thus supports efforts to combat misinformation and fraudulent reviews generated by LLMs (McGovern et al., 2024; Li et al., 2023; Antoun et al., 2023; Lin et al., 2024). Typically, this task is approached as a binary classification problem (Fagni et al., 2021; Jawahar et al., 2020; Mitchell et al., 2023; Lin et al., 2024). Prior studies have explored distinguishing human-written text from LLM-generated text (Ji et al., 2024; Clark et al., 2021), while others have focused on detecting and attributing outputs from different LLMs (Li et al., 2023; McGovern et al., 2024). The latter, also known as Model Attribution (MA) (Antoun et al., 2023), is considerably more challenging due to the similarities in model architectures and training corpus across LLMs.

In this work, we provide the first study of the MA problem in the context of a new class of large language models, namely discrete diffusion large language models (dLLMs) (Nie et al., 2025; Ye et al., 2025; Yu et al., 2025). Unlike autoregressive generation (OpenAI , 2024; Gemini Team,

2025; DeepSeek-AI, 2025), dLLMs treat generation as an iterative decoding process over discrete token sequences (Yu et al., 2025). At each decoding step, any position of the token sequence may be unmasked from a mask token into a concrete token. This decoding process naturally forms a trajectory and introduces dependencies across decoding steps. It also captures, in a finer-grained manner, the distinctive characteristics that different models exhibit when responding to the same text prompt. We show that such unique properties of dLLM decoding trajectories offer a novel basis for addressing the model attribution problem.

Specifically, to fully exploit the fine-grained structural information embedded in the decoding process, we identify two fundamental problems: *how to effectively extract information from the decoding trajectory, and how to utilize such information for model attribution.* To solve these problems, we firstly introduce the *Directed Decoding Map (DDM)* for information extraction. The core motivation behind DDM is that different models exhibit stable differences in how newly decoded tokens interact with previously decoded tokens during generation. By encoding whether a newly decoded token induces positive, negative, or mixed effects on the confidence of previously decoded tokens, as well as the direction of confidence changes for each decoded tokens, DDM transforms complex probabilistic dynamics into a structured representation. This representation reliably captures model-specific decoding characteristics and thereby provides a reliable basis for model attribution. Building on this, we propose a novel model attribution method named *Gaussian-Trajectory Attribution (GTA)*. In GTA, each model is queried with a local dataset to obtain its own collection of DDMs. Based on these DDMs, we fit cell-wise Gaussian distributions separately for each model. This procedure preserves the structural information encoded in DDMs and produces a compact probabilistic fingerprint of each model's decoding behavior. To attribute a target response, we compute the log-likelihood of its DDM under all the constructed model-specific distributions and assign it to the model with the highest likelihood.

We conduct extensive experiments under various settings, including distinguishing between different models as well as between different checkpoints or backups of the same model. We also investigate the influence of different token length and decoding strategies in dLLMs, along with ablation studies on both DDM and GTA. The results consistently demonstrate the strong capability of our method for attributing dLLMs. We summarize our contributions as follows:

- We present the first exploration of model attribution for dLLMs and introduce the use of their distinctive decoding trajectories for lightweight yet reliable attribution. To this end, we design an information extraction scheme named *Directed Decoding Map (DDM)*, which reliably captures the structural and dependency information embedded in the decoding trajectory for a better attribution process.

- We propose a model attribution method named *Gaussian-Trajectory Attribution (GTA)*, which builds compact probabilistic fingerprints for each model via cell-wise Gaussian fitting and attributes a target response to the model with the highest likelihood.

- Extensive experiments across different dLLMs and various attribution settings clearly demonstrate the superiority of DDM and GTA. For example, even in the highly restrictive case where two models are fine-tuned from the same checkpoint using identical configurations, the attribution AUC remains above 81%.

## 2 RELATED WORKS

### 2.1 DISCRETE DIFFUSION LARGE LANGUAGE MODELS (DLLMS)

Discrete Diffusion Large Language Models (dLLMs) (Nie et al., 2025; Ye et al., 2025; Yu et al., 2025) recently emerges as a promising paradigm for non-autoregressive (non-AR) language modeling. In tasks such as code generation (DeepMind, 2025; Inception Labs, 2025), planning (Ye et al., 2025), and Sudoku (Ye et al., 2025), dLLMs have been widely shown to achieve better performance than AR models. In contrast to AR generation, dLLMs treat generation as an iterative decoding process over discrete token sequences (Nie et al., 2025; Ye et al., 2025). This paradigm removes the left-to-right constraint, allowing parallel and structurally controllable generation with bidirectional attention. Some recent works also explore deeply into the scaling-up nature and decoding behavior of dLLMs Yang et al. (2025); Cheng et al. (2025).

Due to the bidirectional nature of dLLMs, their iterative decoding process can be naturally viewed as a trajectory with strong structural information and contextual dependency. The use of historical information from the decoding process has also inspired some current efforts. For instance, DIJA (Wen et al., 2025) and PAD (Zhang et al., 2025) reformulates conventional jailbreak prompts into an interleaved mask-text format, compelling the model to generate unsafe outputs while maintaining contextual consistency. Another work (Xie et al., 2025) shows that dLLMs are more vulnerable to manipulation at the middle of the response than at the initial tokens and proposes MOSA, a reinforcement learning alignment strategy, which requires the model's generated middle tokens to align with a set of predefined safe tokens. In this work, we make the first attempt to use the history trajectory for model attribution. Rather than modifying it as in prior work, we regard the trajectory as a holistic signal and extract from it a representation that highlights model-specific information.

## 2.2 Authorship Attribution

Authorship attribution is a long-standing problem that was initially applied to distinguishing between human authors, with practical applications such as criminal investigations (Chaski, 2005; Koppel et al., 2008; Grant, 2020) and counterterrorism efforts (Cafiero & Camps, 2023; Budryk, 2019). With the rise of large language models (LLMs), and inspired by the Turing Test (Turing, 2007; Biever, 2023; Mei et al., 2024), research has expanded to distinguishing human-written from machine-generated text (Lin et al., 2024). Furthermore, model attribution (MA) further extends the concept of authorship attribution to the machine–machine setting (Li et al., 2023; Lin et al., 2024; Antoun et al., 2023), where the objective is to identify which specific model, or even which version of a model, was responsible for generating a given text.

Existing model attribution methods can be broadly grouped into three categories: statistical feature-based methods (Sharma et al., 2018; Sari et al., 2017; Shrestha et al., 2017; Proisl et al., 2018), which rely on measures such as perplexity, n-grams, and entropy and are lightweight but often limited in performance; classifier-based methods (Solorio et al., 2011; Shao et al., 2019), which train discriminative models on texts from different sources and achieve higher accuracy but at the cost of efficiency and robustness to adversarial mimicking; and watermark-based methods (Kirchenbauer et al., 2023; Boenisch, 2021), which take a pre-hoc approach by embedding watermarks in the generation process and later verifying attribution by detecting these watermarks. In this work, we design a lightweight and reliable statistical feature-based method, for the first time leveraging the unique decoding trajectory of dLLMs to achieve effective model attribution.

## 3 Methods

In this section, we first introduce our proposed information extraction scheme. The core intuition is that relying solely on first-order signals such as model confidence is insufficient to capture the structural information and cross-step dependencies embedded in the decoding trajectories of dLLMs. To address this, we propose a second-order representation, the *Directed Decoding Map (DDM)* in Section 3.1, which explicitly encodes the interdependencies among tokens and steps during decoding into a structured representation. Building on this, we further present our attribution method, *Gaussian-Trajectory Attribution (GTA)* in Section 3.2, which fully leverages the information in DDMs by fitting cell-wise Gaussian distributions over the extracted trajectories, thereby obtaining a compact probabilistic fingerprint of each model's decoding behavior and enabling lightweight yet reliable model attribution.

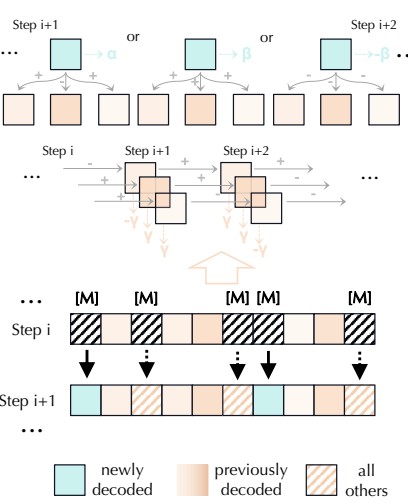

Figure 1: The DDM construction pipeline. [M] is the mask token.

## 3.1 Directed Decoding Map Construction

Consider a dLLM decoding process of $T$ steps producing a sequence of $L$ tokens. We define $c_i(j)$ as the con-

fidence of position $j$ at step $i$ (if $j$ is not yet decoded at step $i$, then $c_i(j) = \varnothing$). The confidence change of a position in two consecutive steps is defined as $\Delta c(j) = c_{i+1}(j) - c_i(j)$.

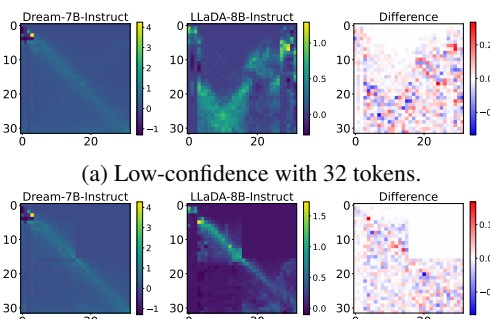

(a) Low-confidence with 32 tokens.

(b) Semi-supervised with 2 blocks and 32 tokens.

Figure 2: DDMs under two decoding strategies: *low-confidence decoding* and *semi-supervised decoding*. The models are instruction-tuned on GSM8K (Cobbe et al., 2021).

Let $U_i = \{j \mid c_i(j) \neq \varnothing\}$ be the set of positions already decoded in step $i$, and $N_{i+1} = U_{i+1} \setminus U_i$ denotes the set of token positions newly decoded at step $i + 1$. We define the Directed Decoding Map entry at the $i$-th row, $j$-th position as $E_i(j)$. Each position falls into one of three categories: newly decoded positions $E_i(n)$, previously decoded positions $E_i(p)$, and all other positions $E_i(o)$. As a starting point, the values in the first row $E_1(j)$ are set to 0.

As illustrated in Figure 1, for each newly decoded position $n \in N_{i+1}$, the change it induces on the confidence of previously decoded positions $p \in U_i$ may exhibit three possible patterns: (i) all $\Delta c(p)$ are non-negative (confidence consistently increases), (ii) all $\Delta c(p)$ are non-positive (confidence consistently decreases), or (iii) a mixture of increases and decreases occurs. We define two distinct effect values $\alpha, \beta \in \mathbb{R}^+$, $\alpha \neq \beta$, and assign the effect value for token $n \in N_{i+1}$ as

$$E_{i+1}(n) = \begin{cases} \alpha, & \exists p, p' \in U_i : \Delta c(p) > 0, \ \Delta c(p') < 0, \\ \beta, & \forall p \in U_i : \Delta c(p) \geq 0, \\ -\beta, & \forall p \in U_i : \Delta c(p) \leq 0. \end{cases} \tag{1}$$

This design captures whether the effect of a new token is mixed, purely positive, or purely negative. For each previously decoded token $p$, once it has been decoded, subsequent steps may either reinforce its confidence or diminish it. Hence, for $p \in U_i$, we assign

$$E_{i+1}(p) = \begin{cases} \gamma, & \Delta c(p) > 0, \\ -\gamma, & \Delta c(p) < 0, \end{cases} \tag{2}$$

where $\gamma \in \mathbf{R}^+, \gamma \neq \alpha \neq \beta$ is another effect value. This rule explicitly encodes whether the stability of an already decoded token is being strengthened or weakened at step $i + 1$. Finally, positions that remain masked carry no effect signal in this step. Thus, for any $o \notin U_{i+1}$, we set $E_i(o) = 0$. The final DDM is represented as a matrix $E \in \mathbb{R}^{T \times L}$, where each entry $E_i(j)$ reflects not only the decoding state but also the directionality of token-level influence across steps. Figure 2 shows DDMs for attribution between two models under two decoding strategies (Nie et al., 2025). For a fair comparison, both models (LLaDA-7B-Instruct (Nie et al., 2025) and Dream-8B-Instruct (Ye et al., 2025)) are instruction-tuned on GSM8K (Cobbe et al., 2021) with identical configurations, and the token number is set to 32. The effect values $\alpha, \beta, \gamma$ are set to 10, 0.5, 2, respectively (we report an ablation study on the effect values in Section 4, which shows that the actual effect value does not substantially affect the performance). As shown, even for different models under the same decoding strategy, DDMs can successfully capture structural differences between them, providing a reliable signal for attribution.

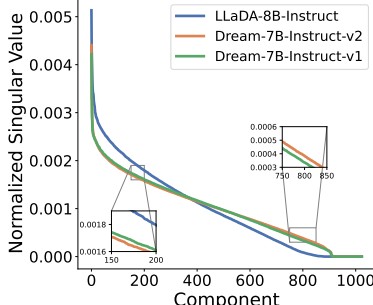

Figure 3: A SVD analysis on the structural information of DDMs.

## 3.2 GAUSSIAN-TRAJECTORY ATTRIBUTION

While the Directed Decoding Map (DDM) provides a structured representation of inter-step dependency, a central challenge remains: *how to effectively preserve and utilize these signals for attribution across models*. To better illustrate the importance of preserving the structural information in DDMs, we perform a SVD analysis (Golub & Van Loan, 2013) (more details

are given in Appendix A.3). Specifically, DDMs are flattened, concatenated into a matrix, mean-centered across features, and then decomposed via SVD. The resulting singular value spectrum characterizes how the variance of DDMs distributes across principal components. In Figure 3, the two versions of Dream-7B-Instruct are tuned from the same checkpoint under identical training configurations. As can be observed, different models exhibit almost identical spectra on the leading components, indicating that they share similar task-level and model-agnostic structures. However, clear discrepancies emerge in the middle and tail components, which correspond to low-energy yet highly discriminative directions. This observation highlights that *attribution signals are primarily encoded in the fine-grained structural patterns rather than in the dominant modes.* Consequently, applying

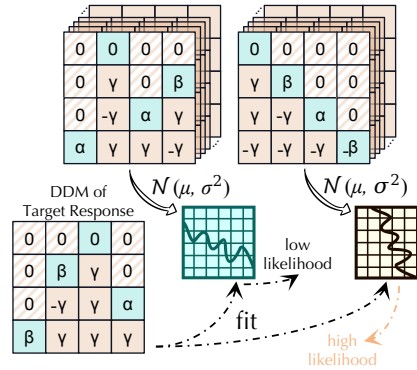

Figure 4: The GTA construction and attribution pipeline.

dimensionality-reduction methods or highly noisy methods that retain only the leading components would inevitably discard these critical differences and significantly weaken attribution performance. To this end, we propose *Gaussian-Trajectory Attribution (GTA)*. By fitting Gaussian distributions to the cell-wise values of the DDM, GTA helps to obtain a compact probabilistic fingerprint that captures both local and global variation, enabling reliable model attribution.

Let a decoding trajectory be represented as a DDM matrix $E \in \mathbb{R}^{T \times L}$, where $T$ is the number of decoding steps and $L$ is the output token length. As illustrated in Figure 4, for each target model $M$, we query it with a local dataset to obtain $N$ trajectories $\{E^{(n)}\}_{n=1}^{N}$. These trajectories are then used to fit an independent Gaussian distribution for each cell $(t, l)$ by computing the empirical mean and variance across samples:

$$\mu_M(t,l) \ = \ \frac{1}{N} \sum_{n=1}^{N} E_{t,l}^{(n)}, \qquad \sigma_M^2(t,l) \ = \ \frac{1}{N} \sum_{n=1}^{N} \left( E_{t,l}^{(n)} - \mu_M(t,l) \right)^2. \tag{3}$$

After the cell-wise Gaussian construction, given a DDM of target response $E^* \in \mathbb{R}^{T \times L}$, its log-likelihood under model $M$ can be formalized as:

$$\ell_M(E^*) \ = \ -\tfrac{1}{2} \sum_{t=1}^{T} \sum_{j=1}^{L} \left[ \frac{\left( E_{t,l}^* - \mu_M(t,l) \right)^2}{\sigma_M(t,l)^2} + \log\left( 2\pi(\sigma_M(t,l)^2) \right) \right]. \tag{4}$$

For a candidate set of target models $\{M_1, \ldots, M_K\}$, we compute the log-likelihood scores $\ell_{M_k}(E^*)$ for each $M_k$. The attribution decision is then made by comparing likelihoods:

$$\hat{M}(E^*) = \arg\max_{M_k} \ \ell_{M_k}(E^*), \tag{5}$$

i.e., we attribute the trajectory to the model under which it achieves the highest likelihood. GTA is lightweight yet powerful: (i) it preserves the structural information encoded by the DDM without collapsing it into coarse statistics, (ii) it produces compact probabilistic fingerprints that are easily comparable across models, and (iii) it enables fine-grained attribution even among models with similar architectures or training corpora.

## 4 EXPERIMENTS

### 4.1 EXPERIMENTAL SETUP

**Models and Datasets.** In our experiments, we consider LLaDA-8B-Instruct (Nie et al., 2025) and Dream-7B-Instruct (Ye et al., 2025), along with multiple variants constructed under different attribution setups. For dLLMs, direct attribution across distinct models is relatively straightforward due to their inherently different decoding strategies. To establish a fairer and more challenging comparison, we additionally instruction-tune both models on identical datasets with the same training configuration (details are given in Appendix A.4). For tuning and evaluation, we use GSM8K (Cobbe et al.,

2021) for math reasoning and CodeAlpaca-20K (Chaudhary, 2023) for code generation, following the LLaDA training pipeline (Nie et al., 2025). Each dataset is split 7:3, with the larger part for training and the smaller for the local dataset. We adopt two decoding strategy: *Low-confidence* decoding and *Semi-supervised* decoding (Nie et al., 2025; Ye et al., 2025), and ensure that each model under attribution uses the same strategy. We use ⟨#tokens, block size⟩ to denote the combination of token length and block size. We consider five combinations: ⟨32, 32⟩ and ⟨64, 64⟩ for Low-confidence decoding and ⟨32, 16⟩, ⟨64, 32⟩, and ⟨64, 16⟩ for Semi-supervised decoding.

Table 1: Attribution Results under the three different setups. Semi-supervised decoding (Nie et al., 2025) is used as the decoding strategy here, where the token length is set to be 32 and the block size is 16. Within each method, the best-performing information representation scheme is underlined, while the column-wise best results are highlighted in green .

| Scenario | Method | Information | GSM8K | | | | CodeAlpaca-20K | | | |
|---|---|---|---|---|---|---|---|---|---|---|
| | | | AUC | TPR@5%FPR | TPR@1%FPR | Acc. | AUC | TPR@5%FPR | TPR@1%FPR | Acc. |
| CMA | Perplexity | | 69.52 | 16.73 | 11.11 | 72.08 | 69.66 | 23.16 | 11.06 | 65.37 |
| | Clustering | confidence | 70.68 | 16.41 | 4.10 | 67.22 | 70.11 | 33.94 | 16.30 | 67.40 |
| | | filtered confidence | 69.68 | 21.01 | 3.52 | 64.83 | 63.33 | 12.75 | 2.07 | 61.58 |
| | | DDM | 97.34 | 94.38 | 90.37 | 95.47 | 97.83 | 92.88 | 71.53 | 94.12 |
| | Distance | confidence | 99.38 | 98.22 | 87.87 | 96.94 | 89.81 | 33.54 | 13.50 | 84.53 |
| | | filtered confidence | 99.85 | 99.91 | 98.66 | 98.91 | 89.59 | 42.73 | 8.44 | 82.60 |
| | | DDM | 99.92 | 99.96 | 98.93 | 99.04 | 97.47 | 84.98 | 64.19 | 92.35 |
| | GTA | confidence | 99.43 | 99.38 | 99.33 | 99.44 | 96.45 | 86.07 | 35.04 | 91.85 |
| | | filtered confidence | 99.66 | 99.02 | 95.67 | 97.99 | 95.66 | 84.22 | 65.11 | 90.35 |
| | | DDM | 99.95 (↑ 30.43) | 99.96 | 99.85 | 99.84 | 98.94 (↑ 35.61) | 98.22 | 92.03 | 97.15 |
| IRA | Perplexity | | 51.58 | 1.69 | 0.45 | 56.09 | 41.72 | 2.16 | 0.57 | 50.02 |
| | Clustering | confidence | 52.23 | 6.24 | 1.52 | 52.70 | 50.60 | 4.62 | 0.67 | 51.06 |
| | | filtered confidence | 60.22 | 7.23 | 1.92 | 59.10 | 51.84 | 12.59 | 6.64 | 51.59 |
| | | DDM | 61.15 | 7.00 | 1.38 | 59.48 | 53.82 | 6.71 | 2.16 | 52.93 |
| | Distance | confidence | 68.31 | 20.03 | 5.40 | 62.85 | 54.98 | 8.27 | 2.29 | 54.22 |
| | | filtered confidence | 68.91 | 22.84 | 7.00 | 64.27 | 55.18 | 6.27 | 1.07 | 54.64 |
| | | DDM | 79.49 | 33.05 | 19.18 | 72.26 | 58.73 | 10.72 | 1.24 | 57.01 |
| | GTA | confidence | 76.79 | 22.30 | 3.79 | 69.13 | 58.31 | 6.34 | 2.39 | 57.28 |
| | | filtered confidence | 80.35 | 31.76 | 5.17 | 73.26 | 54.98 | 8.27 | 2.29 | 54.22 |
| | | DDM | 81.75 (↑ 30.37) | 38.22 | 3.84 | 74.00 | 65.05 (↑ 23.33) | 14.35 | 3.72 | 60.88 |
| CCA | Perplexity | | 49.32 | 6.33 | 1.20 | 52.03 | 40.89 | 4.09 | 0.78 | 50.54 |
| | Clustering | confidence | 49.90 | 3.08 | 0.58 | 52.12 | 51.06 | 4.70 | 0.83 | 51.29 |
| | | filtered confidence | 51.03 | 6.47 | 1.61 | 52.05 | 51.03 | 4.57 | 0.70 | 51.14 |
| | | DDM | 56.33 | 7.63 | 1.56 | 54.57 | 53.79 | 7.51 | 1.94 | 53.49 |
| | Distance | confidence | 61.79 | 5.71 | 1.83 | 59.03 | 59.36 | 5.83 | 1.41 | 60.84 |
| | | filtered confidence | 57.34 | 6.74 | 1.56 | 56.47 | 54.68 | 9.53 | 2.39 | 54.46 |
| | | DDM | 65.84 | 8.61 | 1.43 | 61.24 | 59.47 | 11.15 | 3.05 | 61.14 |
| | GTA | confidence | 64.50 | 13.60 | 3.70 | 60.79 | 53.86 | 5.83 | 1.04 | 53.19 |
| | | filtered confidence | 59.53 | 8.83 | 1.87 | 58.14 | 54.13 | 6.03 | 1.02 | 53.36 |
| | | DDM | 66.91 (↑ 17.59) | 14.50 | 4.68 | 64.92 | 62.64 (↑ 21.75) | 6.38 | 1.31 | 61.78 |

**Attribution Scenario.** We consider three challenging yet important attribution scenarios: **(i)** across different models (e.g., LLaDA-8B-Instruct vs. Dream-7B-Instruct), referred to as Cross-Model Attribution (CMA); **(ii)** between independent runs of the same model initialized from the same checkpoint and tuned using identical training configuration, referred to as Independent-Run Attribution (IRA); and **(iii)** across checkpoints of the same training trajectory at different epochs, referred to as Cross-Checkpoint Attribution (CCA). In our CCA setting, one model is fully trained for 20 epochs, while the other is the checkpoint saved halfway through training (at 10 epochs). Results under smaller intervals are given in Appendix A.8.

**Baseline Methods.** We setup several baseline methods that can be fairly compared in our setting. For information extraction, *Confidence* directly uses the model's predicted probabilities over all positions at each decoding step, without distinguishing between decoded and masked tokens. *Filtered Confidence*, in contrast, leverages the decoded tokens and mask out unfinished positions, thereby retaining only the confidence scores of tokens that have been actually generated. For attribution method, *Clustering* applies unsupervised clustering (in our case, we use DBSCAN (Schubert et al., 2017) with Euclidean distance, an epsilon of 0.8, and a minimum of 20 points to form a cluster) over the trajectory features of responses from different models, and then uses cluster proximity to assign attributing labels. *Distance* computes the Euclidean distances between each target response and the average representations of each model, and uses the relative distance margins as the attri-

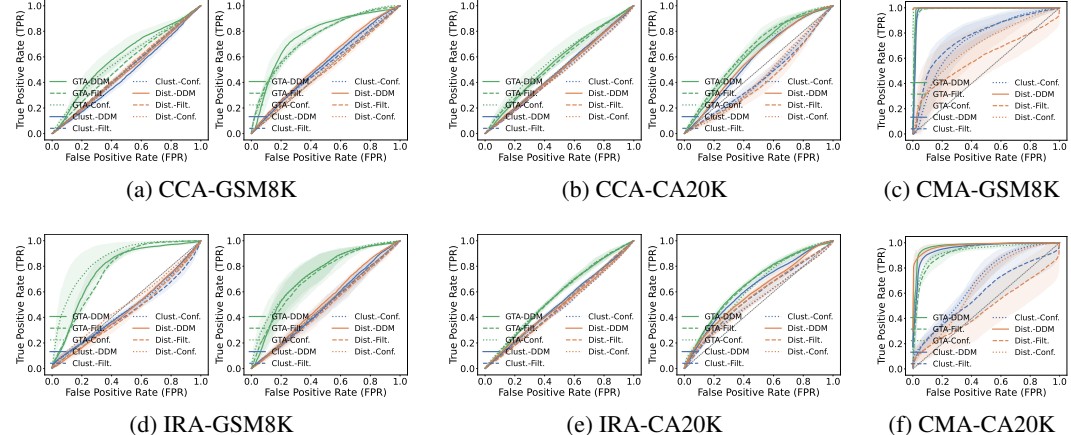

(a) CCA-GSM8K      (b) CCA-CA20K      (c) CMA-GSM8K

(d) IRA-GSM8K      (e) IRA-CA20K      (f) CMA-CA20K

Figure 5: ROC curves of different attribution method–information extraction scheme combinations. *CA20K* abbreviates CodeAlpaca-20K. *Conf.*, *Filt.*, and *Clust.* stand for confidence, filtered confidence, and clustering. Attribution methods are distinguished by color, and extraction schemes by line style. Results are averaged over token lengths and decoding strategies. In subfigures (a)–(d), results on LLaDA are presented on the left, while those on Dream are on the right.

bution score. *Perplexity (PPL)* (Alon & Kamfonas, 2023; Ankner et al., 2024) further aggregates the predicted token probabilities into a score by computing the exponential of the average negative log-likelihood over the response.

**Evaluation Metric.** We adopt AUC score, TPR@Low% FPR, and Accuracy (Acc.) as evaluation metrics. AUC provides a holistic view of performance across different thresholds and serves as a key measure of attribution performance. TPR@Low% FPR highlights performance under stringent false-positive control, which is particularly important in attribution tasks where incorrect attributions can incur high costs. Accuracy offers an intuitive measure of overall correctness and complements of each method.

## 4.2 Main Results

We first provide a global overview of the performance trends across different combinations of attribution methods and information extraction schemes. The results are summarized in Table 1, where we follow prior work by formulating the problem as a binary classification task, i.e., attributing between two models. The token length is set to 32 and the block size to 16. Several key observations can be drawn from the results: **(i).** Among the three attribution scenarios, CMA proves to be the easiest, while CCA is the most challenging. This aligns with intuition: differences in decoding behavior across different models (CMA) are generally more pronounced than those between checkpoints or backups of the same model. Moreover, in CCA, the compared models can be regarded as one model being further fine-tuned from the other, whereas in IRA the compared models share the same initialization and undergo a fine-tuning process with same configuration. Consequently, the inter-model gap in CCA is usually smaller than that in IRA. **(ii).** Across different methods, DDM consistently demonstrates superior performance, yielding roughly a 10% AUC improvement over others under almost all settings. **(iii).** GTA achieves consistently stronger attribution performance than alternative methods, with the combination of GTA and DDM (i.e., our attribution method) delivering the best results. In all settings, this method improves AUC by 20–30% compared to PPL.

In addition, to provide a more intuitive comparison, we report the ROC curves of different methods under various settings. Results are given in Figure 5, where *CA20K* denotes the abbreviation of CodeAlpaca-20K. *Conf.*, *Filt.*, and *Clust.* denote confidence, filtered confidence, and clustering, respectively. Each attribution method is represented by a distinct color, while each information extraction scheme is indicated by a specific line style. Within subfigures (a), (b), (d) and (e), the left panel shows the results under LLaDA, while the right panel corresponds to Dream. The shaded regions indicate the variance across five token length and decoding strategy settings we consider (see Section 4.1), As can be observed, all three information extraction schemes under GTA outperform the baselines, and DDM consistently enables more stable attribution. Specifically, in the more chal-

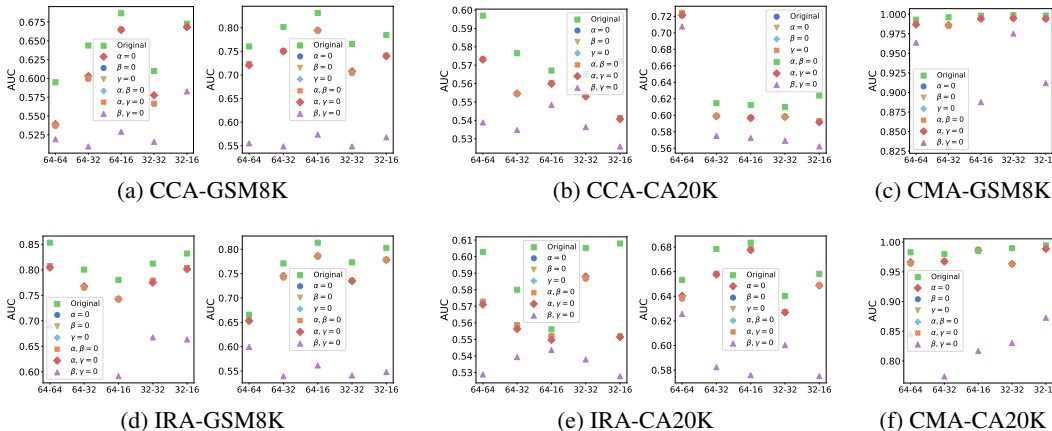

(a) CCA-GSM8K        (b) CCA-CA20K        (c) CMA-GSM8K

(d) IRA-GSM8K        (e) IRA-CA20K        (f) CMA-CA20K

Figure 6: This experiment reveals two key aspects: (i) the impact of zeroing out specific effect values, and (ii) the effect of decoding strategy and token length on performance. In subfigures (a)–(d), results on LLaDA are presented on the left, while those on Dream are on the right.

lenging CCA and IRA settings, nearly all baselines perform close to random guessing. In contrast, for the relatively easier CMA setting, some baselines achieve moderate AUC but remain suboptimal; only a few methods combined with DDM, such as Dist.-DDM on GSM8K and Dist.-DDM / Clust.-DDM on CodeAlpaca-20K, attain strong performance. These results highlight the robustness and effectiveness of DDM and GTA across diverse scenarios.

## 4.3 ABLATION STUDY

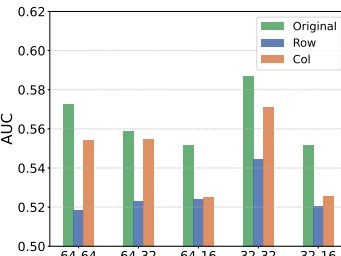

Figure 7: Maintaining the structural integrity of DDMs is critical for reliable attribution.

**Structural Information of DDM.** We conduct an experiment to investigate the importance of the structural information extracted by DDM from decoding trajectories. Specifically, we zero out each of the three effect values individually and in pairs, resulting in six different variants. As shown in Figure 6, we evaluate these variants along with the original DDM (denoted as Original) under five combinations of token length and decoding strategy, with different colors representing different variants.

Several insights can be drawn: (i) among the three effect values, $\gamma$ plays the most critical role, followed by $\beta$. In particular, when both $\beta$ and $\gamma$ are zeroed out, the performance drops drastically. In other cases, performance still degrades but the decline is less severe and relatively consistent. $\gamma$ corresponds to the trajectory of the confidence variations of previously decoded tokens $E_i(p)$ at each decoding step, while $\beta$ represents the trajectory of the one-way influence exerted by the newly decoded token $E_i(n)$ on $E_i(p)$.

The substantial contribution of these two components indicates that DDM's effectiveness relies heavily on modeling both the historical semantic accumulation of tokens (captured by $\gamma$) and the interaction between newly decoded and previously decoded tokens (captured by $\beta$). In other words, DDM goes beyond local confidence signals by leveraging structural and dependancy information from the decoding trajectory, which is key to its superior attribution performance.

**Influence of Decoding Strategy and Token Length.** Besides the influence of effect values, Figure 6 also illustrates the impact of different decoding strategies and token lengths. Focusing on the results of the original DDM (green square), we observe that for CMA (Figures 6c, 6f), these factors

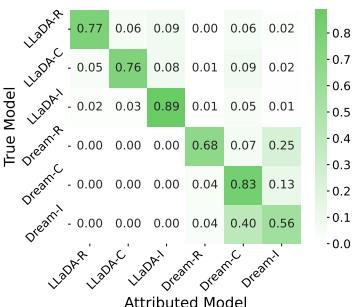

Figure 8: Attribution across multiple models conducted within a single attribution process.

have little effect on performance. In contrast, for CCA (Figures 6a, 6b) and IRA (Figures 6d, 6e), performance varies slightly across settings and no consistent pattern emerges, with AUC changes generally within 0.1. Considering that token length and decoding strategies can be flexibly chosen in practice, and that our experiments show their influence on performance to be small, the feasibility of applying DDM in attribution is thus well supported.

**Structure Preservation of GTA.** A primary motivation behind the design of GTA is to better preserve the structural information in DDMs. To assess the necessity and impact of our design, we conduct a comparative study in Figure 7, where the cell-wise Gaussian distribution in GTA is replaced with token-wise (column-based, denoted as Col) and step-wise (row-based, denoted as Row) variants. The results show that when structural information is disrupted, performance drops significantly, particularly in step-wise Gaussian (Row), where the interdependence among tokens is severely undermined, leading to a partial failure of attribution. This further highlights the utility of GTA's design.

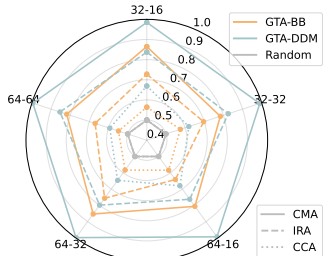

Figure 9: The performance of GTA under black-box setting.

**Attribution across Multiple Models.** Although model attribution is typically formulated as a binary classification task, performing attribution across multiple models simultaneously provides a stronger validation of a method's scalability. In Figure 8, we conduct attribution over six model variants used in this work and report the accuracy. We use GSM8K as the dataset in this experiment, with the token length and decoding strategy set to 64 and 32, respectively. Here, *-R* denotes the reference model, *-C* the model compared with *-R* in CCA, and *-I* the model compared with *-R* in IRA. The diagonal entries indicate cases where the predicted model matches the ground-truth model, i.e., attribution is correct. We observe that in most cases, the combination of GTA and DDM consistently achieve reliable attribution, and the limited deviations that arise are predominantly restricted to models within the same family.

**Influence of Effect Values.** Besides the default value of $\alpha$, $\beta$ and $\gamma$ (10, 0.5, 2), we also experiment with other values to demonstrate the robustness of our method to such variations. As shown in Table 2, we vary the three effect values ($\alpha \in [5, 15]$ step 1; $\beta \in [0.1, 1.0]$ step 0.1; $\gamma \in [1.5, 2.5]$ step 0.1) and report the AUC std (%) with token length 64 on GSM8K under low-confidence decoding. The std values are mostly below 0.1% and at most around 1%, indicating that DDM and GTA is robust to changes in effect values. More results regarding the influence of effect values can be found in Appendix A.5.

Table 2: Performance variations under different effect values.

| std (%) | $\nabla$CMA | $\nabla$IRA | $\nabla$CCA |
|---|---|---|---|
| $\alpha_{5\sim15}$ | 0.08 | 1.67 | 1.49 |
| $\beta_{0.1\sim1.0}$ | 0.02 | 0.09 | 0.08 |
| $\gamma_{1.5\sim2.5}$ | 0.03 | 0.86 | 0.60 |

**GTA in Black-box Scenario.** Previous confidence-based methods, including DDM, are operated in the *gray-box* setting, where the adversary lacks access to model parameters or intermediate features and can only rely on model outputs. We further push this boundary by evaluating GTA in the most strict scenario, namely the full *black-box* setting, where the adversary can only observe the final decoded tokens at each step without any confidence information. In this case, GTA constructs distributions directly from the model's decoding history. Results are reported in Figure 9, where GTA-BB refers to GTA under black-box setting. As shown, while the performance degrades, it remains considerable. This demonstrates that for dLLMs, the attribution problem is still solvable even under the strictest setting, with their unique decoding mechanism playing a key role in enabling this.

Table 3: Cross-domain attribution results of GTA on GSM8K.

| Information | Setting | AUC | TPR@1%FPR |
|---|---|---|---|
| confidence | In.D | 64.50 | 3.70 |
| | Crs.D | 59.27 | 1.60 |
| filtered confidence | In.D | 59.53 | 1.87 |
| | Crs.D | 56.90 | 1.53 |
| **DDM (ours)** | In.D | **66.91** | **4.68** |
| | Crs.D | **62.38** | **3.12** |

**Generalizability of GTA.** We evaluate the cross-domain robustness of our method, i.e., how well it performs when the target dataset differs from the one used for model training. In this setting, we use GSM8K for model training and CodeAlpaca-20K is used as the attribution target. For ease of comparison, we denote the default in-domain setting as In.D and

the cross-domain setting as Crs.D. Since most other attribution methods fail to operate reliably under the cross-domain scenario, we report only the results of different combinations within GTA. Other dataset configuration and training/attribution pipeline is the same as in Table 1.

As can be observed in Table 3, the cross-domain setting is clearly more challenging. While performance does decrease, the drop remains modest, typically within 5%. This indicates that DDM-GTA exhibits strong resilience under cross-domain conditions.

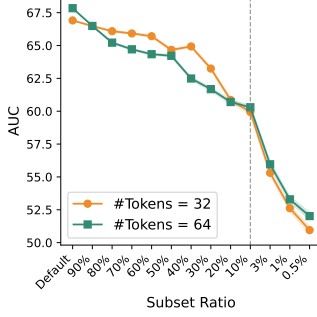

Figure 10: Effect of dataset scale for constructing DDM.

**Influence of data scale for DDM construction.** We conduct an experiment on GSM8K to evaluate how the size of the dataset used to construct the DDM affects attribution performance. Specifically, we randomly subsample the training data at 90%, 80%, ..., 10%, 3%, 1%, and 0.5% of the default dataset size and do the DDM construction process under each subset. Each subsampling scale is repeated five times. We report results for #tokens = 32 and 64, with the block size fixed to 16. As can be observed in Figure 10, when using 10% or more of the data, the AUC consistently remains above approximately 60%. Only in the most extreme setting, i.e., using merely 0.5% of the data, does the performance degrade to the level of random guessing. These results indicate that DDM-GTA has modest data requirements and remains effective even with substantially reduced data.

## 5 CONCLUSION

In this work, we take the first step toward exploring the problem of model attribution in dLLMs. As a new class of large language models, dLLMs exhibit distinctive decoding characteristics. Building on the decoding trajectories of dLLMs, we propose Directed Decoding Map (DDM), an information extraction scheme that captures the interdependencies among tokens across decoding steps. The strong structural properties of DDMs provide a clear reflection of model-specific behaviors. Furthermore, we introduce Gaussian Trajectory Attribution (GTA), which constructs cell-wise Gaussian distributions over DDMs, thereby preserving fine-grained structural information and producing a compact probabilistic fingerprint of each model. Extensive experiments across diverse settings demonstrate the efficacy of our method.

## LLM DISCLAIMER

In this work, Large Language Models (LLMs) are utilized exclusively for non-technical purposes, such as assisting with literature review and refining the readability of the manuscript. Their use was limited to improving phrasing and the presentation. No technical contributions, such as methodological design, model implementation, or experimental analysis, involve the use of LLMs. The authors have full responsibility for the final text.

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

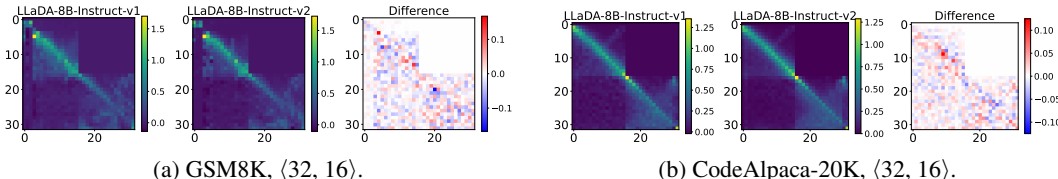

(a) GSM8K, ⟨32, 16⟩.  (b) CodeAlpaca-20K, ⟨32, 16⟩.

Figure 11: DDM comparison of LLaDA-8B-Instruct instruction-tuned under two different datasets. ⟨#tokens, block size⟩ is set to ⟨32, 16⟩.

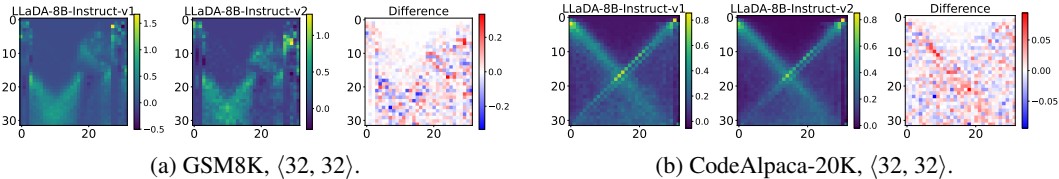

(a) GSM8K, ⟨32, 32⟩.  (b) CodeAlpaca-20K, ⟨32, 32⟩.

Figure 12: DDM comparison of LLaDA-8B-Instruct instruction-tuned under two different datasets. ⟨#tokens, block size⟩ is set to ⟨32, 32⟩.

# A  APPENDIX

## A.1  MORE DDM VISUALIZATIONS.

In Figures 11 12 13 14, we further provide several visualizations of DDMs under IRA setting, with the models to be attributed set to LLaDA-8B-Intruct and datasets used are GSM8K and CodeAlpaca-20K. All the five ⟨#tokens, block size⟩ settings are provided.

## A.2  MORE RESULTS OF MULTIPLE MODEL ATTRIBUTION

In Figure 18, we further provide several results under multiple models attribution. The dataset used here is GSM8K and ⟨#tokens, block size⟩ is set to ⟨32, 16⟩, ⟨32, 32⟩, ⟨64, 16⟩, ⟨64, 64⟩, respectively.

## A.3  ANOTHER SVD ANYLASIS UNDER SEMI-SUPERVISED DECODING.

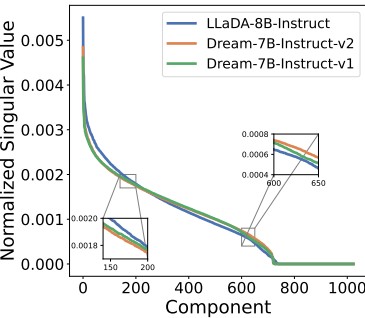

Figure 19: A SVD analysis on the structural information of DDMs.

In the main text, we analyzed the structural information of DDM using SVD under low-confidence decoding (⟨#tokens, block size⟩ = ⟨32, 32⟩). Here, we provide another example under semi-supervised decoding with ⟨#tokens, block size⟩ = ⟨32, 16⟩, as shown in Figure 19. The same phenomenon is observed: different models share almost identical spectra in the leading components, reflecting similar task-level, model-agnostic structures, while clear discrepancies appear in the middle and tail components, which capture low-energy yet highly discriminative directions. This suggests that *attribution signals lie in fine-grained structural patterns rather than dominant modes*, and thus dimensionality reduction or noisy methods that retain only leading components would discard critical differences and weaken attribution.

## A.4  TRAINING CONFIGURATIONS AND PROMPTS USED IN OUR WORK

In Figures 15 and 16, we provide prompt examples for the two datasets used in our work (GSM8K and CodeAlpaca-20K), where GSM8K is for mathematical reasoning task and CodeAlpaca-20K is for code generation task. In Table 4, we report the detailed training configurations of the models used in our paper.

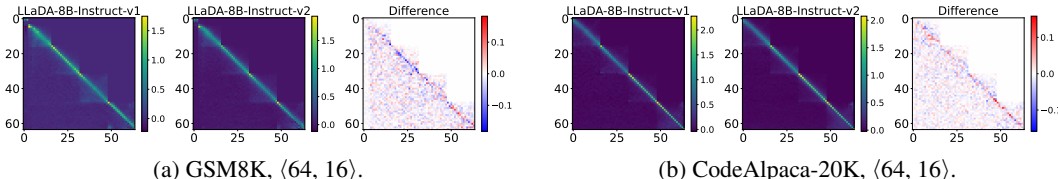

(a) GSM8K, $\langle 64, 16 \rangle$.     (b) CodeAlpaca-20K, $\langle 64, 16 \rangle$.

Figure 13: DDM comparison of LLaDA-8B-Instruct instruction-tuned under two different datasets. $\langle$#tokens, block size$\rangle$ is set to $\langle 64, 16 \rangle$.

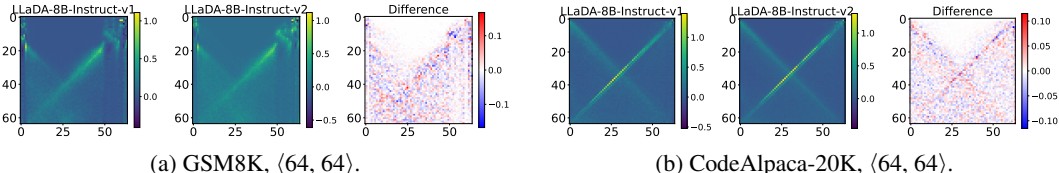

(a) GSM8K, $\langle 64, 64 \rangle$.     (b) CodeAlpaca-20K, $\langle 64, 64 \rangle$.

Figure 14: DDM comparison of LLaDA-8B-Instruct instruction-tuned under two different datasets. $\langle$#tokens, block size$\rangle$ is set to $\langle 64, 64 \rangle$.

Table 5: Key symbols and notations.

| Symbol | Description |
|---|---|
| $T$ | Number of decoding steps |
| $L$ | Length of the generated token sequence |
| $c_i(j)$ | Model confidence of position $j$ at decoding step $i$ |
| $\Delta c(j)$ | Confidence change between steps $i$ and $i+1$ at position $j$ |
| $U_i$ | Set of positions already decoded at step $i$ |
| $N_{i+1}$ | Set of positions newly decoded at step $i+1$ |
| $E_i(j)$ | Value of the Directed Decoding Map (DDM) at step $i$ and position $j$ |
| $E_i(n)$ | DDM value for a newly decoded position at step $i$ |
| $E_i(p)$ | DDM value for a previously decoded position at step $i$ |
| $E_i(o)$ | DDM value for a still-masked position at step $i$ (set to 0) |
| $\alpha, \beta, \gamma$ | Positive effect values ($\alpha, \beta, \gamma \in \mathbb{R}_+$) |
| $E$ | DDM matrix, $E \in \mathbb{R}^{T \times L}$ |
| $M$ | A candidate dLLM in the attribution task |
| $K$ | Number of candidate models considered in the attribution task |
| $\mu_M(t, l)$ | Gaussian mean of cell $(t, l)$ in the DDMs of model $M$ |
| $\sigma_M^2(t, l)$ | Gaussian variance of cell $(t, l)$ in the DDMs of model $M$ |
| $\ell_M(E^*)$ | Log-likelihood of a target DDM $E^*$ under model $M$ in GTA |
| $\widehat{M}(E^*)$ | Attributed model for trajectory $E^*$ |

## A.5   MORE EXPERIMENTS ON THE EFFECT VALUES.

Here, we conduct a deeper investigation into the effect values. To eliminate scale-*related biases*, we sample three distinct values from a same range level to serve as the effect values, and three different range levels are used: (0,10.0], (0,100.0], and (0,1000.0]. For each range level, we perform five random samplings and report the averaged results. We set the token length and block size to 32 and 16, respectively. The results under GSM8K and CodeAlpaca-20K are summarized in Table 6. It can be observed that the choice of the effect value has only a negligible impact on performance. Conceptually, the effect value simply serves as a positional marker for different signal types. Nevertheless, some patterns can still be observed from the results. For example, random sampling of the effect value remains effective, but restricting the sampling range to a smaller interval leads to more stable outcomes and yields slight (though marginal) performance improvements.

| Setting | Value |
|---|---|
| *Training Arguments* | |
| Epochs | 20 |
| Batch size | 1 |
| Gradient accumulation | 4 |
| Logging steps | 2 |
| Max seq. length | 4096 |
| Save steps | 100 |
| Learning rate | 1e-5 |
| Weight decay | 0.1 |
| Max grad norm | 1.0 |
| *Deepspeed Config* | |
| Zero stage | 2 |
| Gradient accumulation | 4 |
| Gradient clipping | 1.0 |
| Zero3 init flag | False |
| Processes | 8 |

Table 4: Training configuration.

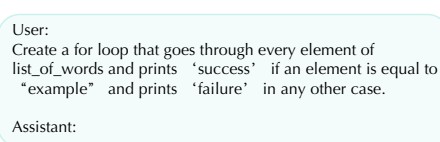

User:
Respond in the following format:
<reasoning>
Your reasoning here
</reasoning>
<answer>
...
</answer>
Stefan goes to a restaurant to eat dinner with his family. They order an appetizer that costs $10 and 4 entrees that are $20 each. If they tip 20% of the total for the waiter, what is the total amount of money that they spend at the restaurant?

Assistant:

Figure 15: Prompt for GSM8K.

User:
Create a for loop that goes through every element of list_of_words and prints 'success' if an element is equal to "example" and prints 'failure' in any other case.

Assistant:

Figure 16: Prompt for CodeAlpaca-20K.

Figure 17: Training configuration (left) and prompt examples (right).

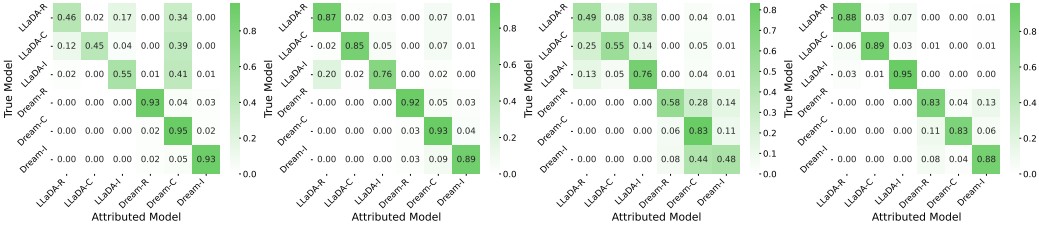

Figure 18: Attribution for multiple models under GSM8K. ⟨#tokens, block size⟩ is set to ⟨32, 16⟩, ⟨32, 32⟩, ⟨64, 16⟩, ⟨64, 64⟩, respectively.

Table 6: Attribution accuracy under different DDM ranges.

| | GSM8K Cobbe et al. (2021) | | | | CodeAlpaca-20K Chaudhary (2023) | | | |
|---|---|---|---|---|---|---|---|---|
| | Default | (0, 10.0] | (0, 100.0] | (0, 1000.0] | Default | (0, 10.0] | (0, 100.0] | (0, 1000.0] |
| CMA | 99.95 | $99.87 \pm \mathbf{0.06}$ | $99.79 \pm \mathbf{0.10}$ | $99.65 \pm \mathbf{0.19}$ | 98.94 | $98.84 \pm \mathbf{0.08}$ | $98.76 \pm \mathbf{0.13}$ | $98.62 \pm \mathbf{0.18}$ |
| IRA | 81.75 | $81.84 \pm \mathbf{0.16}$ | $81.59 \pm \mathbf{0.19}$ | $81.37 \pm \mathbf{0.26}$ | 65.05 | $64.96 \pm \mathbf{0.12}$ | $64.79 \pm \mathbf{0.18}$ | $64.63 \pm \mathbf{0.23}$ |
| CCA | 66.91 | $66.59 \pm \mathbf{0.24}$ | $66.44 \pm \mathbf{0.27}$ | $66.40 \pm \mathbf{0.30}$ | 62.64 | $62.55 \pm \mathbf{0.19}$ | $62.47 \pm \mathbf{0.26}$ | $62.32 \pm \mathbf{0.29}$ |

## A.6 SYMBOLS AND NOTATIONS.

The key symbols and notations used in our work are given in Table 5.

Table 7: Runtime comparison of different attribution methods.

| | GSM8K Cobbe et al. (2021) | | | CodeAlpaca-20K Chaudhary (2023) | |
|---|---|---|---|---|---|
| #Tokens | Method | Time (s) | #Tokens | Method | Time (s) |
| **32** | **GTA (ours)** | 0.38 | **64** | **GTA (ours)** | 0.85 |
| | **Clustering** | 0.49 (∼1.3×) | | **Clustering** | 1.30 (∼1.5×) |
| | **Distance** | 0.73 (∼2×) | | **Distance** | 1.22 (∼1.5×) |
| | **SVD** | 4.53 (∼12×) | | **SVD** | 25.18 (∼30×) |

Table 8: Attribution accuracy under different epoch intervals (for CCA) on GSM8K.

| Method | Information | GSM8K | | | | | | | | | |
| | | 10 (Default) | 9 | 8 | 7 | 6 | 5 | 4 | 3 | 2 | 1 |
|---|---|---|---|---|---|---|---|---|---|---|---|
| **Distance** | confidence | 61.79 | 59.47 | 58.38 | 56.21 | 53.63 | 52.19 | 50.38 | 50.22 | 50.47 | 50.06 |
| | filtered confidence | 57.34 | 56.83 | 55.12 | 54.47 | 53.26 | 53.05 | 51.42 | 51.01 | 50.72 | 50.06 |
| | **DDM** | **65.84** | **63.48** | **61.69** | **60.06** | **58.59** | **56.40** | **54.37** | **53.69** | **53.24** | **51.28** |
| **GTA** | confidence | 64.50 | 63.32 | 61.03 | 59.80 | 57.12 | 54.26 | 52.80 | 52.18 | 51.36 | 50.62 |
| | filtered confidence | 59.53 | 57.44 | 55.63 | 54.38 | 53.49 | 52.64 | 51.47 | 51.07 | 50.98 | 50.54 |
| | **DDM (ours)** | **66.91** | **64.95** | **62.37** | **61.88** | **60.51** | **58.35** | **57.02** | **55.63** | **53.26** | **52.79** |

## A.7    EFFICIENCY OF GTA.

In Table 7, we report the attribution time on the GSM8K target set for all methods. We also include the time required to compute the SVD in this table. We can observe that GTA requires the shortest computation time for attribution and scales well as the number of tokens increases.

## A.8    INFLUENCE OF EPOCH INTERVAL.

In our CCA experiment, one model is fully trained for 20 epochs, while the other is the checkpoint saved halfway through training (at 10 epochs). Here, we conduct an additional experiment where the second model is taken from later checkpoints (at 11, 12, ..., 19 epochs). The token length is set to 32 and the block size is 16. The results under LLaDA are shown in Table 8.

Since perplexity-based and clustering-based approaches already perform poorly under our default setting, we focus on the remaining methods. We can observe from the above table that, as the two models become closer in terms of training stage, all methods exhibit a similar trend of gradually decreasing attribution performance, which is expected given the diminishing divergence between the models. However, the DDM–GTA combination consistently performs the best, maintaining an AUC above 55% even when the interval drops to 3 epochs. Among the other methods, the strongest is the DDM–distance combination, which achieves above 55% AUC down to an interval of 5 epochs.

