# OpenReview forum: "Every Step Counts: Decoding Trajectories as Authorship Fingerprints of dLLMs"
_ICLR.cc/2026/Conference — Submitted to ICLR 2026_

### Official Review · Reviewer_3QFe · 2025-10-25

**Soundness:** 2
**Presentation:** 3
**Contribution:** 2
**Rating:** 4
**Confidence:** 4

**Summary:**

This paper proposes a model attribution method targeting discrete diffusion large language models. Diverging from conventional methods reliant solely on the final text output, this work utilizes the dLLMs' unique non-autoregressive decoding trajectory as a model "fingerprint". By analyzing how the model’s internal states evolve over timesteps during the denoising process (i.e., the decoding trajectory), it is possible to effectively determine the source model responsible for generating a particular response.

However, this innovation is severely limited by a lack of theoretical robustness proof and significant practical deployment challenges.

**Strengths:**

1. Model attribution is an interesting and timely research direction.

2. The experimental results preliminarily demonstrate the effectiveness of the proposed method.

**Weaknesses:**

1. The decoding process of diffusion models is inherently highly stochastic and sampling-dependent. The paper fails to provide clear empirical and theoretical arguments on how the extracted "trajectory fingerprint" maintains its uniqueness and stability under varying critical conditions, such as different random seeds, sampling temperatures, and other decoding hyperparameter settings, which are essential for its qualification as a robust "fingerprint."

2. The proposed method requires logging the entire internal state at every single time step ($T$ steps) during the decoding process. This generates massive storage and computational overhead compared to storing just the final output text, making large-scale practical deployment highly impractical. Additionally, trajectory-based attribution requires access to intermediate outputs during model decoding, which is a typical white-box or gray-box assumption. For commercial closed-source LLM services like OpenAI or Google, users cannot access these decoding trajectories, rendering the method entirely infeasible.

3. A proper comparative analysis is missing. The authors must incorporate, or at least fully discuss the limitations of, established attribution methods for autoregressive models, such as those based on final response log-likelihood or watermarking techniques.

4. The paper only considers two small-scale models: LLaDA-8B-Instruct and Dream-7B-Instruct. The robustness of the approach across larger and more diverse model architectures remains unproven.

5. All key symbols used in the formulas should be listed in a symbol table to aid reader comprehension.

6. The paper appears not to conduct cross-domain generalization tests.

**Questions:**

1. Which part of the decoding trajectory contributes most to attribution? Theoretical analysis should provide insights into the information content of different timesteps to support feature extraction design.

2. What metrics are used to quantify and compare the similarity or difference between two decoding trajectories?

3. The paper abbreviates “Discrete Diffusion Large Language Models” as dLLM, but in other literature, diffusion LLMs may not be explicitly discrete. Why did the authors include “Discrete”? What is the standard terminology, and is the proposed method not applicable in continuous spaces?

---

> ### Author Response · Authors · 2025-11-20
> **Author Response [1/4]**
>
> It’s our great honor to receive Reviewer 3QFe's thoughtful comments and constructive feedback to our work. We would like to address all the questions reflected in the review below.
>
> ---
>
> > **[W1]:** How the extracted trajectory fingerprint maintains its uniqueness and stability under varying critical conditions, e.g., different seeds, temperatures, and other decoding hyperparameters.
>
> Thanks for the valuable comment. For the two most critical hyperparameters in the decoding process, **token length** and **decoding strategy**,  experiments are already conducted in our paper (see **Fig. 5** for details). Here, we perform four further ablation studies as follows:
>
> - Effect of different **random seeds**.
>
> Specifically, we perform model inference using different global random seeds. Besides the default seed of 42, we additionally use three randomly sampled seeds: 781, 59, and 346. The token length set to 32 and block size set to 16. The averaged results across the four runs on LLaDA are reported in the table below.
>
> ||GSM8K |CodeAplaca-20K
> |-|-|-
> |AUC|66.92±0.03|62.61±0.05
> |TPR@5%FPR|14.52±0.04|6.35±0.06
> |TPR@1%FPR|4.65±0.03|1.30±0.02
> |Acc.|64.90±0.05|61.75±0.04
>
> As can be observed, the random seed has **negligible impact** on the attribution performance. This is because, for dLLMs such as LLaDA, it only influences **which specific token is sampled to unmask a given position** during inference, while the **contextual dependencies and the bi-directional decoding nature** of dLLMs effectively suppress such variance. In contrast, hyperparameters such as token length and block size exert a much stronger influence.
>
> - Effect of **temperature**.
>
> We also varied the categorical distribution temperature for LLaDA over 0.0, 0.1, 0.2, 0.3, with the token length set to 32 and block size set to 16. The results are summarized below.
>
> ||GSM8K||||CodeAplaca-20K||||
> |-|-|-|-|-|-|-|-|-
> |**Temperature**|AUC|TPR@5%FPR|TPR@1%FPR|Acc.|AUC|TPR@5%FPR|TPR@1%FPR|Acc.
> |**0.0**|66.91|14.50|4.68|64.92|62.64|6.38|1.31|61.78
> |**0.1**|67.03|14.69|4.57|64.88|62.26|5.89|1.24|61.30
> |**0.2**|66.52|14.21|4.39|64.15|62.38|6.02|1.35|61.26
> |**0.3**|66.74|14.58|4.80|64.35|62.87|6.14|1.29|61.57
>
> We observe that the temperature has only a minor effect, as it merely **adjusts the stochasticity in choosing among the top-confidence tokens**, rather than influencing which tokens have high confidence at each step.
>
> - The **scale of the dataset** used for DDM construction.
>
> We evaluate the impact of dataset size on attribution performance using GSM8K. Specifically, we randomly subsample the training data to 90%,80%,…,10%,3%,1%, and 0.5% of its original size and do the DDM construction under each setting. Each subsampling ratio is repeated five times. Results for 32 and 64 tokens with a fixed block size of 16 are reported below.
>
> |#Tokens|Default|90%|80%|70%|60%|50%|40%|30%|20%|10%|3%|1%|0.5%
> |-|-|-|-|-|-|-|-|-|-|-|-|-|-
> |32|66.91|66.48±0.05|66.09±0.08|65.92±0.06|65.71±0.11|64.67±0.10|64.93±0.08|63.25±0.13|60.86±0.17|59.94±0.24|55.31±0.21|52.62±0.25|50.95±0.28
> |64|67.84|66.49±0.04|65.22±0.08|64.71±0.07|64.34±0.08|64.22±0.09|62.49±0.12|61.68±0.13|60.70±0.16|60.30±0.14|55.96±0.24|53.29±0.22|52.02±0.27
>
> We find that when using **10% or more** of the data, the AUC consistently **remains above approximately 60%**. Only in the most extreme setting, i.e., using **merely 0.5% of the data**, does the performance **degrade to the level of random guessing**. These results indicate that DDM-GTA **has modest data requirements** and remains effective even with substantially reduced data.
>
> - The influence of **the effect values**.
>
> We conduct a deeper investigation into the effect values. To eliminate scale-related biases, we sample three distinct values from a same range level to serve as the effect values, and three different range levels are used: (0,10.0], (0,100.0], and (0,1000.0]. For each range level, we perform five random samplings and report the averaged results. We set the token length and block size to 32 and 16. The results are summarized below:
>
> |_GSM8K_|Default|(0, 10.0]|(0, 100.0]|(0, 1000.0]|_CodeAlpaca-20K_|Default|(0, 10.0]|(0, 100.0]|(0, 1000.0]
> |-|-|-|-|-|-|-|-|-|-
> |CMA|99.95|99.87±0.06|99.79±0.10|99.65±0.19|CMA|98.94|98.84±0.08|98.76±0.13|98.62±0.18
> |IRA|81.75|81.84±0.16|81.59±0.19|81.37±0.26|IRA|65.05|64.96±0.12|64.79±0.18|64.63±0.23
> |CCA|66.91|66.59±0.24|66.44±0.27|66.40±0.30|CCA|62.64|62.55±0.19|62.47±0.26|62.32±0.29
>
> We can observe that the choice of the effect value **has only a negligible impact on performance**. Conceptually, they can be viewed as **positional markers for different signal types**. Nevertheless, some patterns can still be observed. For example, restricting the sampling range to **a smaller interval** leads to more stable outcomes and yields slight (though marginal) performance improvements.
>
> We sincerely hope that our response above helps to address this concern. We would be glad to provide further clarifications if needed.

---

> ### Author Response · Authors · 2025-11-20
> **Author Response [2/4]**
>
> > **[W2]:** Regarding the storage/computational overhead and the gray-box assumption.
>
> Thanks for the valuable comment. We would like to clarify these two points as follows:
>
> - Potential storage and computational overhead:
>
> Each decoding step is an inherent part of the model’s inference process. Thus, the trajectory generated during decoding naturally **exists without additional computation**. Many dLLMs even provide dedicated functions to conveniently record such information during the generation of the model output:
>
> ```
> output = model.diffusion_generate(...)
>
> history = output.history
> ```
>
> The full decoding trajectory can be **directly accessed** with a single line of code as above. Therefore, **our approach introduces no extra storage requirements or computational cost**. The decoding process itself for dLLMs is also **much more time consuming** than the DDM construction process. Moreover, we do not rely on the full token logits across the entire vocabulary at each decoding step. Our DDM only captures the inter-step dependencies and structural relations extracted from adjacent decoding steps, which forms **a single compact matrix representation**. When computing GTA, **we avoid any matrix operations, only cell-wise scalar computations (mean and variance) are involved**. This design keeps the process highly lightweight and eliminates unnecessary computation overhead. For clarity, we also report the attribution time comparison of different methods on the GSM8K target dataset:
>
> |#Tokens|Method|Time (s)|#Tokens|Method|Time (s)|
> |-|-|-|-|-|-|
> |**32**|**GTA** (ours)|0.38|**64**|**GTA** (ours)|0.85|
> ||**Clustering**|0.49 (~1.3x)||**Clustering**|1.30 (~1.5x)|
> ||**Distance**|0.73 (~2x)||**Distance**|1.22 (~1.5x)|
> ||**SVD**|4.53 (~12x)||**SVD**|25.18 (~30x)|
>
> We can observe that GTA requires **the shortest computation time** for attribution and **scales well** as the number of tokens increases.
>
> - The white-box / gray-box setting.
>
> We consider our default setting as a gray-box scenario, where the evaluator can only access the model’s output confidence, but not any internal information such as gradients or model parameters. Such scenario is not only relevant for external evaluators, but also **provide a practical auditing tool for model developers and owners to monitor potential privacy issues in their deployed models**. In addition, **experiments in the black-box setting are conducted in our original paper**, where we can only access the decoded tokens each step. **Please refer to Figure 9 and lines 466–474 for details**.
>
> Concerning **closed-source APIs**, there exits **some commercial APIs such as OpenAI’s GPT-4 expose the top-k (e.g., top-5) per-token probabilities**, which effectively constitutes a gray-box scenario under our definition. Such accessible confidence information can be leveraged to conduct our proposed experiments without requiring direct access to model internals. For **commercial dLLMs**, since this is still an emerging field, **no callable commercial dLLM APIs were available as of the submission deadline (and till now this is still the case)**. Only some web-based chat model like Gemini Diffusion and Mercury Coder are available. Therefore, we believe it is **both necessary and timely to study their potential privacy issues before large-scale deployment**.
>
> ---
>
> > **[W3]:** Regarding the limitations of established attribution methods for autoregressive models, such as those based on final response log-likelihood or watermarking techniques.
>
> Thanks for the comment. We completely agree that this is an important aspect, and as noted in **lines 130–139** of our paper, we have already discussed the relevant existing methods in detail.

---

> ### Author Response · Authors · 2025-11-20
> **Author Response [3/4]**
>
> > **[W4]:** The robustness of the approach across larger and more diverse model architectures.
>
> Thanks for the valuable suggestion. We would like to clarify that, **as of the submission deadline, no dLLMs larger than the ones we used had been publicly released**. To meet your requirement, here we additionally conduct experiments on **SDAR-30B-A3B-Chat** [1] and **dLLM-Var** [2], which were released on October 7 and October 28, respectively.
>
> The dataset configuration and training/attribution pipeline is the same as in the paper. We set the token length and block size to 32 and 16, respectively. For SDAR-30B-A3B-Chat, we adopt the static decoding strategy. For dLLM-Var, we use the default decoding setup. Results under the **most challenging CCA setting** are given below:
>
> - **SDAR-30B-A3B-Chat** [1]:
>
> ||||GSM8K||||CodeAlpaca-20K||||
> |-|-|-|-|-|-|-|-|-|-|-|
> |Scenario|Method|Information|AUC|TPR@5%FPR|TPR@1%FPR|Acc.|AUC|TPR@5%FPR|TPR@1%FPR|Acc.|
> |CCA|Perplexity|Perplexity|51.23|5.27|1.35|52.79|49.93|4.54|1.03|51.46|
> ||Clustering|confidence|52.31|5.42|1.48|52.79|52.38|5.21|2.59|52.51|
> |||filtered confidence|52.15|6.83|1.92|53.03|51.97|4.89|2.46|52.49|
> |||**DDM**|**55.27**|**8.27**|**2.68**|**57.56**|**54.78**|**7.98**|**3.12**|**54.06**|
> ||Distance|confidence|58.86|4.76|1.37|56.54|57.78|6.52|2.40|57.51|
> ||| filtered confidence|57.69|3.52|1.62|54.91|56.83|4.93|2.84|56.82|
> ||| **DDM**|**63.81**|**9.31**|**2.33**|**60.83**|**60.28**|**12.36**|**5.68**|**58.84**|
> ||**GTA**|confidence|61.59|15.72|3.65|60.46|55.41|6.73|3.64|54.08|
> |||filtered confidence|59.13|13.52|2.86|59.31|54.52|7.32|5.34|54.04|
> |||**DDM** (ours)|**65.98**|**17.65**|**4.38**|**64.24**|**62.87**|**12.59**|**7.03**|**61.68**|
>
> - **dLLM-Var** [2]:
>
> ||||GSM8K||||CodeAlpaca-20K||||
> |-|-|-|-|-|-|-|-|-|-|-|
> |Scenario|Method|Information|AUC|TPR@5%FPR|TPR@1%FPR|Acc.|AUC|TPR@5%FPR|TPR@1%FPR|Acc.|
> |CCA|Perplexity|Perplexity|51.47|3.68|1.67|51.65|48.65|4.65|1.31|51.06|
> ||Clustering|confidence|51.42|4.46|0.92|50.97|51.65|4.28|1.09|51.84|
> |||filtered confidence|52.54|6.62|1.32|52.08|51.20|3.86|1.68|51.49|
> |||**DDM**|**56.89**|**7.68**|**1.95**|**56.06**|**55.39**|**5.63**|**1.64**|**54.40**|
> ||Distance|confidence|57.45|3.85|1.06|56.37|58.62|4.58|1.45|58.25|
> ||| filtered confidence|59.65|3.40|0.96|58.92|58.65|3.53|1.34|58.04|
> ||| **DDM**|**62.96**|**8.32**|**2.73**|**61.65**|**61.75**|**10.98**|**4.95**|**60.13**|
> ||**GTA**|confidence|60.73|11.48|2.86|60.32|59.41|5.86|2.96|59.10|
> |||filtered confidence|61.66|12.68|4.05|60.08|60.56|6.30|2.58|59.36|
> |||**DDM** (ours)|**67.06**|**15.84**|**6.26**|**65.28**|**65.25**|**11.06**|**5.38**|**63.50**|
>
> As can be observed, DDM together with GTA **remains the most useful** attribution method on these two models. Also, when DDM is cooperated with other attribution methods, it can still help to improve the performance.  Besides, we can observe that for SDAR-30B-A3B-Chat, the attribution performance is particularly strong in the **low-FPR regime (TPR@Low FPR)**. For example, the GPA-DDM combination achieves over 7% TPR@1% FPR on the CodeAlpaca-20K dataset. This indicates that the model retains **highly discriminative attribution signals** even under extremely low false-positive constraints.
>
> We sincerely hope that our response can alleviate this concern. We would be more than happy to address any further questions you may have regarding this point.
>
> [1] SDAR: A Synergistic Diffusion-AutoRegression Paradigm for Scalable Sequence Generation
>
> [2] Diffusion LLM with Native Variable Generation Lengths: Let [EOS] Lead the Way
>
> ---
>
> > **[W5]:** All key symbols used in the formulas should be listed in a symbol table to aid reader comprehension.
>
> Thanks for the helpful suggestion. A symbol table summarizing the main notation used in the formulas is as follows:
>
> |Symbol|Description|
> |-|-|
> |$T$|Number of decoding steps
> |$L$|Length of the generated token sequence
> |$c_i(j)$|Model confidence of position $j$ at decoding step $i$
> |$\Delta c(j)$|Confidence change between steps $i$ and $i+1$ at position $j$
> |$U_i$|Set of positions already decoded at step $i$
> |$N_{i+1}$|Set of positions newly decoded at step $i+1$
> |$E_i(j)$|Value of the Directed Decoding Map (DDM) at step $i$ and position $j$
> |$E_i(n)$|DDM value for a newly decoded position at step $i$
> |$E_i(p)$|DDM value for a previously decoded position at step $i$
> |$E_i(o)$|DDM value for a still-masked position at step $i$ (set to $0$)
> |$\alpha,\beta,\gamma$|Positive effect values ($\alpha, \beta, \gamma \in \mathbb{R}_+$)
> |$E$|DDM matrix, $E \in \mathbb{R}^{T \times L}$
> |$M$|A candidate dLLM in the attribution task
> |$K$|Number of candidate models considered in the attribution task
> |$\mu_M(t,l)$|Gaussian mean of cell $(t,l)$ in the DDMs of model $M$
> |$\sigma_M^2(t,l)$|Gaussian variance of cell $(t,l)$ in the DDMs of model $M$
> |$\ell_M(E^*)$|Log-likelihood of a target DDM $E^*$ under model $M$ in Gaussian-Trajectory Attribution (GTA)
> |$\widehat{M}(E^\*)$|Attributed model for trajectory $E^\*$

---

> ### Author Response · Authors · 2025-11-20
> **Author Response [4/4]**
>
> > **[W6]:** Conduct cross-domain generalization tests.
>
> Thanks for highlighting this important consideration. Here, we evaluate our method  in a cross-domain scenario, i.e., how well it performs when the target dataset differs from the one used for model training. In this setting, we use GSM8K (CodeAlpaca-20K) for model training and CodeAlpaca-20K (GSM8K) is used as the attribution target. We denote the default in-domain setting as _In.D_ and the cross-domain setting as _Crs.D_. Since most other methods fail to operate reliably under this scenario, we report only the results of different combinations within GTA. Other dataset configuration and training/attribution pipeline is the same as in the paper. The results are given below:
>
> |||GSM8K||||CodeAplaca-20K||||
> |-|-|-|-|-|-|-|-|-|-|
> |Information|Setting|AUC|TPR@5%FPR|TPR@1%FPR|Acc.|AUC|TPR@5%FPR|TPR@1%FPR|Acc.|
> |confidence|In.D|**64.50**|**13.60**|**3.70**|**60.79**|**53.86**|**5.83**|**1.04**|**53.19**|
> ||Crs.D|59.27|9.20|1.60|57.29|52.84|3.40|0.60|52.17|
> |filtered confidence|In.D|**59.53**|**8.83**|**1.87**|**58.14**|**54.13**|**6.03**|**1.02**|**53.36**|
> ||Crs.D|56.90|6.68|1.53|54.45|52.78|4.46|1.25|52.60|
> |**DDM** (ours)|In.D|**66.91**|**14.50**|**4.68**|**64.92**|**62.64**|**6.38**|**1.31**|**61.78**|
> ||Crs.D|62.38|12.78|3.12|60.16|58.32|5.97|1.13|57.39|
>
> As can be observed, the cross-domain setting is more challenging, but the performance drop remains modest, typically within 5%. This indicates that DDM-GTA exhibits strong resilience under cross-domain conditions.
>
> ---
>
> > **[Q1]:** Which part of the decoding trajectory contributes most to attribution?
>
> Thanks for the interesting question. We believe it is difficult to pinpoint precisely, since different models may **adopt different decoding methods** (e.g., low-confidence, semi-supervised, etc.). Moreover, **as the decoding length, training stage, and data vary**, it becomes challenging to conclude which specific decoding step or token position contributes most to the model’s final output. Therefore, our analysis focuses on **the decoding process as a whole**. As mentioned in the paper, our experiments reveal that **the structural information and token dependencies** within the decoding trajectory play a crucial role in attribution. To effectively extract such information, we leverage the historical records stored during decoding, focusing on **how the confidence of unmasked tokens changes across adjacent steps**. In this way, we can capture essential structural dependencies while filtering out redundant information. Meanwhile, it remains **computationally lightweight**: involving only **scalar operations** on token confidence between steps and **avoiding heavy matrix operations and dimensionality reduction*.
>
> ---
>
> > **[Q2]:** What metrics are used to quantify and compare the similarity or difference between two decoding trajectories?
>
> Thanks for the question. We use the **log-likelihood** to quantify similarity. Detailed explanations are provided in **Eqs. (3)–(5) and lines 236–259**.
>
> ---
>
> > **[Q3]:** The paper abbreviates “Discrete Diffusion Large Language Models” as dLLM, but in other literature, diffusion LLMs may not be explicitly discrete. Why did the authors include “Discrete”? What is the standard terminology, and is the proposed method not applicable in continuous spaces?
>
> Thanks for the important question. _Discrete Diffusion Large Language Models (dLLMs)_ represent a new class of models that are **distinct from** both _Continuous Diffusion Large Language Models_ and _Autoregressive-based LLMs_. For your kind reference, two survey papers provide detailed discussions: one specifically focuses on **discrete diffusion** LLMs [1], while the other provides a broader overview covering **both discrete and continuous diffusion** LLMs [2].
>
> Regarding the abbreviation _dLLM_, it has become a relatively common term in this field. With that said, **there are indeed several naming variations in different works**: some works refer to these models as "diffusion-based LLMs" and abbreviate them as _dLLM_ [3,6,7] or _DLM_ [5], while others adopt _DDLM_ as shorthand for "Discrete Diffusion Language Models" [4]. Given this inconsistency in terminology, we chose one of the most widely used and recognizable conventions in current practice.
>
> [1] Discrete Diffusion in Large Language and Multimodal Models: A Survey
>
> [2] A survey on diffusion language models
>
> [3] The devil behind the mask: An emergent safety vulnerability of diffusion llms
>
> [4] Planner and Executor: Collaboration between Discrete Diffusion And Autoregressive Models in Reasoning
>
> [5]Attention is all you need for kv cache in diffusion llms
>
> [6] d^2Cache: Accelerating Diffusion-Based LLMs via Dual Adaptive Caching
>
> [7] Mask Tokens as Prophet: Fine-Grained Cache Eviction for Efficient dLLM Inference
>
> ---
>
> Thanks again for the thoughtful and constructive feedback. If reviewer 3QFe has any remaining concerns, we would definitely love to clarify further.

---

> ### Author Response · Authors · 2025-11-27
>
> Dear Reviewer 3QFe,
>
> Thanks again for your thoughtful comments and constructive feedback to our work. As the discussion period is coming to a close soon, we would very appreciate it if you could let us know if our response has sufficiently addressed your concerns, or if there are any remaining questions we can clarify.
>
> Also, based on your valuable comments, we've add relevant experiments and discussions into our revised manuscript. If anything remains unclear or unsatisfactory, we would also be glad to refine the manuscript further.

---

### Official Review · Reviewer_dGL7 · 2025-10-30

**Soundness:** 2
**Presentation:** 3
**Contribution:** 3
**Rating:** 6
**Confidence:** 4

**Summary:**

This paper addresses the model attribution problem in dLLMs by proposing DDM to extract decoded trajectory structure information and GTA to achieve attribution through Gaussian distribution. It performs well in multi-scenario experiments across models and is superior to the baseline. It is also effective in black-box scenarios, thus helping to manage the risks of dLLMs.

**Strengths:**

- This paper presents the first study on model attribution in DLLMs.

- This work is experimentally comprehensive, investigating various scenarios, such as different numbers of models and black-box states.

**Weaknesses:**

- Lack of analysis explaining why GTA can capture structural patterns rather than dominant modes.

- Lack of a detailed definition or calculation formula for confidence.

- The authors mention the high efficiency of GTA, but lack specific experimental data.

- For Cross-Checkpoint Attribution (CCA), the paper does not provide specific numerical values ​​for the epoch interval, nor does it experimentally compare the minimum epoch interval that different methods can measure.

**Questions:**

- What do the values ​​in Figure 8 mean? Are they model similarity? If so, what is the accuracy of the model attribution?

- Why not directly compare the SVD of the DDM, but instead use a log-likelihood to compare similarity?

---

> ### Author Response · Authors · 2025-11-20
> **Author Response [1/2]**
>
> We would like to express our sincere appreciation for reviewer dGL7’s valuable comments and support for our work. We respond to the questions as follows.
>
> ---
>
> > **[W1]:** Explaining why GTA can capture structural patterns rather than dominant modes.
>
> Thanks for the valuable suggestion. This can be mainly attributed to two factors. First, building on **the bi-directional decoding nature** of dLLMs, the **DDM construction** captures useful cross-step dependencies and structural information in the decoding process, which helps reduce information redundancy and preserve informative signals for GTA. Second, GTA operates **without lossy compression**, allowing these structural information across tokens and steps to be retained. Together, these aspects contribute to the effectiveness of GTA in capturing structural patterns. We'll make the discussion clearer in the revision.
>
> ---
>
> > **[W2]:** Lack of a detailed definition or calculation formula for confidence.
>
> Thanks for the kind note. For the calculation of the confidence, we follows the default pipeline in LLaDA: during each decoding step, the model first produces **logits for all positions**, which are converted into **token predictions $x_0$** via argmax sampling (optionally with Gumbel noise). The confidence for each position is then computed as **the predicted probability of the selected token**, obtained by applying a softmax over the logits and gathering the probability corresponding to  $x_0$. Only **positions currently masked** contribute valid confidence values, all others are assigned **$-\infty$**. These confidence scores are then used to rank positions for selecting tokens to fill the mask, yielding a step-by-step confidence trajectory throughout the diffusion decoding process. We will add the above explanation into the revision to make things clearer.
>
> ---
>
> > **[W3]:** Specific experimental data regarding the efficiency of GTA.
>
> Thanks for the valuable comment. In the table below, we report the attribution time on the GSM8K target set for all methods. To facilitate answering your another question later, we also include the time required to compute the SVD in this table.
>
> |#Tokens|Method|Time (s)|#Tokens|Method|Time (s)|
> |-|-|-|-|-|-|
> |**32**|**GTA** (ours)|0.38|**64**|**GTA** (ours)|0.85|
> ||**Clustering**|0.49 (~1.3x)||**Clustering**|1.30 (~1.5x)|
> ||**Distance**|0.73 (~2x)||**Distance**|1.22 (~1.5x)|
> ||**SVD**|4.53 (~12x)||**SVD**|25.18 (~30x)|
>
> We can observe that GTA requires **the shortest computation time** for attribution and **scales well** as the number of tokens increases. We'll add the results above in the revision.

---

> ### Author Response · Authors · 2025-11-20
> **Author Response [2/2]**
>
> > **[W4]:** For Cross-Checkpoint Attribution (CCA), the paper does not Provide the specific numerical values for the epoch interval in Cross-Checkpoint Attribution (CCA), and experimentally compare the minimum epoch interval that different methods can measure.
>
> Thanks for the thoughtful comment. In our CCA experiment, one model is **fully trained for 20 epochs**, while the other is **the checkpoint saved halfway through training (at 10 epochs)**. We will clarify this setup in the revision. We also **agree with the reviewer** that comparing the minimum epoch interval required for successful attribution is meaningful. To this end, we conduct an additional experiment where the second model is **taken from later checkpoints (at 11, 12, …, 19 epochs)**. The token length is set to 32 and the block size is 16. The results under LLaDA are shown in the table below.
>
> ||_GSM8K_|Epoch Interval||||||||||_CodeAplaca-20K_|Epoch Interval||||||||||
> |-|-|-|-|-|-|-|-|-|-|-|-|-|-|-|-|-|-|-|-|-|-|-|
> | Method |Information|10 (Default)|9|8|7|6|5|4|3|2|1| Information |10 (Default)|9|8|7|6|5|4|3|2|1|
> |**Distance**|confidence|61.79|59.47|58.38|56.21|53.63|52.19|50.38|50.22|50.47|50.06|confidence|59.36|58.36|57.42|56.58|55.28|54.33|53.49|51.22|50.12|50.02|
> ||filtered confidence|57.34|56.83|55.12|54.47|53.26|53.05|51.42|51.01|50.72|50.06|filtered confidence|54.68|54.33|53.26|52.84|52.07|51.60|51.33|50.76|50.40|50.12|
> ||**DDM**|**65.84**|**63.48**|**61.69**|**60.06**|**58.59**|**56.40**|**54.37**|**53.69**|**53.24**|**51.28**|**DDM**|**59.47**|**58.92**|**58.12**|**57.63**|**55.96**|**55.38**|**54.26**|**53.12**|**52.25**|**50.37**|
> |**GTA**|confidence|64.50|63.32|61.03|59.80|57.12|54.26|52.80|52.18|51.36|50.62|confidence|53.86|53.17|52.65|52.08|51.66|51.12|50.73|50.76|50.72|50.40|
> ||filtered confidence|59.53|57.44|55.63|54.38|53.49|52.64|51.47|51.07|50.98|50.54|filtered confidence|54.13|53.58|53.01|52.65|52.20|51.75|51.23|50.86|50.34|50.12|
> ||**DDM** (ours)|**66.91**|**64.95**|**62.37**|**61.88**|**60.51**|**58.35**|**57.02**|**55.63**|**53.26**|**52.79**|**DDM** (ours)|**62.64**|**61.78**|**61.24**|**60.43**|**58.97**|**57.42**|**56.23**|**55.42**|**53.82**|**51.30**|
>
> Since perplexity-based and clustering-based approaches already perform poorly under our default setting, we focus on the remaining methods. We can observe from the above table that, as the two models become closer in terms of training stage, all methods exhibit **a similar trend of gradually decreasing attribution performance**, which is expected given the diminishing divergence between the models. However, the DDM–GTA combination consistently performs the best, maintaining an AUC above 55% even when the interval drops to 3 epochs. Among the other methods, the strongest is the DDM–distance combination, which achieves above 55% AUC down to an interval of 5 epochs. We'll add the above results and discussions in the revision.
>
> ---
>
> > **[Q1]:** What do the values in Figure 8 mean? Are they model similarity? If so, what is the accuracy of the model attribution?
>
> Thanks for the valuable question. In Figure 8, what we intend to report is the **attribution accuracy** when performing attribution across multiple models. Each **row** corresponds to the **ground-truth model**, and each **column** corresponds to the **attributed model**. The diagonal entries represent the ground-truth accuracy (i.e., the proportion of correctly attributed data among all target samples of that ground-truth model). The values in each row sum to 1. We will make the description clearer in the revision.
>
> ---
>
> > **[Q2]:** Why not directly compare the SVD of the DDM, but instead use a log-likelihood to compare similarity?
>
> Thanks for the insightful question. The primary reason is related to **efficiency**. As can be observed in the table in our response to your **[W3]**, computing the SVD involves several **matrix operations** that are not very efficient, and **the computational cost scales poorly** as the number of tokens increases. In contrast, GTA performs cell-wise computations, which are considerably **more lightweight and scalable**. Besides, one secondary reason is that, using SVD-based approaches would unnecessarily **introduce additional hyperparameters**. For example, whether to use a spectral distance in the singular-value space, a principal-angle distance between subspaces, a combined metric, or whether optimal rotational alignment should be applied. Given these considerations, we chose the simpler yet more efficient design of GTA to do the attribution.
>
> ---
>
> Thanks again for the constructive feedback and support for our work. If reviewer dGL7 has any remaining concerns, we are happy to clarify further.

---

> > ### Comment · Reviewer_dGL7 · 2025-11-28
> >
> > Thank you for the detailed responses regarding my questions. I'm satisfied with the your response. I have also read other reviews and responses and can see that the authors have done their sincere best to respond to issues raised by my fellow reviewers.

---

> ### Author Response · Authors · 2025-11-27
>
> Dear Reviewer dGL7,
>
> Thanks again for your valuable comments and strong support to our work. As the discussion period is coming to a close soon, we would very appreciate it if you could let us know if our response has sufficiently addressed your concerns, or if there are any remaining questions we can clarify.
>
> Also, following your valuable suggestions, we have incorporated the relevant experiments and discussions into the revised manuscript. If there is anything that is still unclear or unsatisfactory, we would be very happy to make additional improvements.

---

### Official Review · Reviewer_gzXx · 2025-11-01

**Soundness:** 2
**Presentation:** 3
**Contribution:** 3
**Rating:** 4
**Confidence:** 2

**Summary:**

The paper studies model attribution for discrete diffusion large language models (dLLMs) by exploiting their iterative decoding trajectories. It proposes the Directed Decoding Map (DDM), a second-order representation that encodes how newly decoded tokens affect previously decoded tokens, and Gaussian-Trajectory Attribution (GTA), which fits cell-wise Gaussian fingerprints over DDMs for attribution scoring. Experiments across cross-model, independent-run, and cross-checkpoint settings report strong AUCs—especially for GTA+DDM—compared to perplexity, clustering, and distance baselines. The paper argues that discriminative signal lives in fine-grained (tail) components (SVD analysis) and that results are robust to effect-value choices and decoding strategies.

**Strengths:**

- Clear, novel framing: leverages unique bidirectional, iterative decoding of dLLMs to form attribution features (DDMs) rather than raw confidences.

- Method is simple, lightweight, and gray-/black-box compatible (cell-wise Gaussians; black-box variant uses only decoded tokens).

- Strong empirical gains over PPL/cluster/distance baselines across scenarios; thorough ROC/AUC reporting.

- Insightful SVD analyses showing signal in tail components; ablations on effect values and structure preservation.

**Weaknesses:**

- Independence assumption: GTA models each cell independently; correlations across steps/tokens appear important per SVD/structure results, yet are not modeled (risking suboptimality). Please justify and/or compare to structured densities (e.g., low-rank Gaussians).

- Definition of “confidence” $c_i(j)$ is underspecified (distribution over what, pre/post-masking calibration, temperature), affecting DDM entries. Clarify computation and normalization across steps/models.

- Effect values $(\alpha,\beta,\gamma)$ are fixed hyperparameters; although robustness is claimed, the choice (e.g., $\alpha=10,\beta=0.5,\gamma=2$) seems ad hoc and may encode scale information from confidences. Provide calibration/normalization or learning-based selection.

**Questions:**

- [How exactly is $c_i(j)$ computed (logits→probabilities? temperature? masking convention)?

- Why independent cell Gaussians? Did you try low-rank covariance, Kronecker or HMM-like temporal models to capture step/token dependencies?

-  How many samples N per model are needed to fit stable GTA fingerprints? Provide AUC vs N and confidence intervals.

- Does performance hold under domain shift (e.g., train GTA on GSM8K, test attribution on CodeAlpaca) and under adversarial imitation of DDM patterns?

---

> ### Author Response · Authors · 2025-11-20
> **Author Response [1/3]**
>
> It is our great honor to receive reviewer gzXx's valuable comments and encouraging feedback. We would like to respond to the questions as follows.
>
> ---
>
> > **[W1, Q2]:** GTA models each cell independently and correlations across steps/tokens appear important. Justify and/or compare to structured densities (e.g., low-rank Gaussians).
>
> Thanks for the valuable comment. We would like to clarify that the correlations across steps and tokens, specifically, the cross-step dependencies as well as the structural information among tokens, are **explicitly encoded in the DDMs before using GTA**. As shown in Figure 7, our experiments suggest that these correlations play an important role in attribution performance, which **aligns with the intuition behind your kind comment**.
>
> Besides that, following your suggestion, we additionally construct a low-rank Gaussian baseline. Specifically, for each model, after collecting the DDMs, we flatten them into feature vectors and fit a low-rank Gaussian distribution:
>
> $$
> p(\mathbf{x}) = \mathcal{N}(\boldsymbol{\mu}, U U^\top + \sigma^2 I)
> $$
>
> The matrix $U \in \mathbb{R}^{d \times r}$ captures the dominant subspace (obtained via SVD with a 95% energy threshold), and $\sigma^2 I$ models the isotropic residual noise. During attribution, given a new model response, we compute its log-likelihood under each model-specific low-rank Gaussian using the Woodbury identity:
>
> $$
> \log p(\mathbf{x}) = -\tfrac{1}{2}\left[(\mathbf{x}-\boldsymbol{\mu})^\top \Sigma^{-1} (\mathbf{x}-\boldsymbol{\mu}) + \log|\Sigma| + d \log(2\pi)\right]
> $$
>
> which allows **efficient estimation** of the structured density. This setup provides a fair comparison between our method and a standard form of structured probabilistic modeling. The corresponding results are summarized in the table below:
>
> ||GSM8K||||CodeAplaca-20K||||
> |-|-|-|-|-|-|-|-|-|
> |Method|AUC|TPR@5%FPR|TPR@1%FPR|Acc.|AUC|TPR@5%FPR|TPR@1%FPR|Acc.|
> |**GTA-DDM** (ours)|**66.91**|**14.50**|**4.68**|**64.92**|**62.64**|**6.38**|**1.31**|**61.78**|
> |**Low-rank Gaussian**|63.84 (-3.07)|12.92 (-1.58)|3.26 (-1.42)|62.38 (-2.54)|58.26 (-4.38)|5.69 (-0.69)|1.07 (-0.24)|57.72 (-4.06)|
>
> Empirically, we find that the low-rank Gaussians underperform our method, the reason behind is that it only captures the **coarse structural correlations**, but fails in distinguishing fine-grained, step-dependent variations crucial for faithful attribution, confirming the necessity of GTA’s design. We also report the attribution time on the GSM8K target set for both methods in the table below.
>
> |#Tokens|Method|Time (s)|#Tokens|Method|Time (s)|
> |-|-|-|-|-|-|
> |32|**GTA-DDM** (ours)|**0.38**|64|**GTA-DDM** (ours)|**0.85**|
> ||Low-rank Gaussian|3.68 (~10x)||Low-rank Gaussian|21.79 (~25x)|
>
> As shown, DDM-GTA is lightweight and scalable, whereas the low-rank Gaussian baseline is **noticeably less efficient** (although implemented with an efficient estimation). We sincerely hope that our response can alleviate this concern. Please feel free to let us know if any additional clarification is needed, we would be happy to address it.
>
> ---
>
> > **[W2, Q1]:** Definition of “confidence” ci(j) is underspecified.
>
> Thanks for the kind note. For the calculation of the confidence, we follows the default pipeline in LLaDA: during each decoding step, the model first produces **logits for all positions**, which are converted into **token predictions $x_0$** via argmax sampling with Gumbel noise. The confidence for each position is then computed as **the predicted probability of the selected token**, obtained by applying a softmax over the logits and gathering the probability corresponding to  $x_0$. Only **positions currently masked** contribute valid confidence values, all others are assigned **$-\infty$**. These confidence scores are then used to rank positions for selecting tokens to fill the mask, yielding a step-by-step confidence trajectory throughout the diffusion decoding process. We will add the above explanation into the revision to make things clearer.

---

> ### Author Response · Authors · 2025-11-20
> **Author Response [2/3]**
>
> > **[W3]:** Although robustness is claimed, the choice of effect values seems ad hoc and may encode scale information from confidences.
>
> Thanks for the valuable suggestion. We further conduct a deeper investigation into the effect values. To eliminate **scale-related biases**, we **sample three distinct values from a same range level** to serve as the effect values, and three different range levels are used: (0,10.0], (0,100.0], and (0,1000.0]. For each range level, we perform five random samplings and report the averaged results. We set the token length and block size to 32 and 16, respectively. The results under GSM8K and CodeAlpaca-20K are summarized below:
>
> |_GSM8K_|Default|(0, 10.0]|(0, 100.0]|(0, 1000.0]|_CodeAlpaca-20K_|Default|(0, 10.0]|(0, 100.0]|(0, 1000.0]|
> |-|-|-|-|-|-|-|-|-|-|
> |CMA|99.95|99.87±0.06|99.79±0.10|99.65±0.19|CMA|98.94|98.84±0.08|98.76±0.13|98.62±0.18|
> |IRA|81.75|81.84±0.16|81.59±0.19|81.37±0.26|IRA|65.05|64.96±0.12|64.79±0.18|64.63±0.23|
> |CCA|66.91|66.59±0.24|66.44±0.27|66.40±0.30|CCA|62.64|62.55±0.19|62.47±0.26|62.32±0.29|
>
> It can be observed that the choice of the effect value **has only a negligible impact on performance**. Conceptually, the effect value simply serves as a **positional marker for different signal types**. Nevertheless, some patterns can still be observed from the results. For example, random sampling of the effect value remains effective, but restricting the sampling range to **a smaller interval** leads to more stable outcomes and yields slight (though marginal) performance improvements. We will include the corresponding results and discussion in the revision.
>
> ---
>
> > **[Q3]:** How many samples N per model are needed to fit stable GTA fingerprints? Provide AUC vs N and confidence intervals.
>
> Thanks for the insightful comment. We conduct an experiment on GSM8K to evaluate how the size of the dataset used to construct the DDM affects attribution performance. Specifically, we randomly subsample the training data at 90%, 80%, …, 10%, 3%, 1%, and 0.5% of the default dataset size and do the DDM construction process under each subset. Each subsampling scale is repeated five times. We report results for #tokens = 32 and 64, with the block size fixed to 16. The results are summarized in the table below.
>
> |#Tokens|Default|90%|80%|70%|60%|50%|40%|30%|20%|10%|3%|1%|0.5%|
> |-|-|-|-|-|-|-|-|-|-|-|-|-|-|
> |32|66.91|66.48±0.05|66.09±0.08|65.92±0.06|65.71±0.11|64.67±0.10|64.93±0.08|63.25±0.13|60.86±0.17|59.94±0.24|55.31±0.21|52.62±0.25|50.95±0.28|
> |64|67.84|66.49±0.04|65.22±0.08|64.71±0.07|64.34±0.08|64.22±0.09|62.49±0.12|61.68±0.13|60.70±0.16|60.30±0.14|55.96±0.24|53.29±0.22|52.02±0.27|
>
> We find that when using **10% or more** of the data, the AUC consistently **remains above approximately 60%**. Only in the most extreme setting, i.e., using **merely 0.5% of the data**, does the performance **degrade to the level of random guessing**. These results indicate that DDM-GTA **has modest data requirements** and remains effective even with substantially reduced data. We will include the corresponding results and discussion in the revision.

---

> ### Author Response · Authors · 2025-11-20
> **Author Response [3/3]**
>
> > **[Q4]:** Does performance hold under domain shift (e.g., train GTA on GSM8K, test attribution on CodeAlpaca) and under adversarial imitation of DDM patterns?
>
> Thanks for the valuable comment. We conduct two experiments in respond to this concern:
>
> - For domain shift, we evaluate the cross-domain robustness of our method, i.e., how well it performs when the target dataset differs from the one used for model training. In this setting, we use GSM8K (CodeAlpaca-20K) for model training and CodeAlpaca-20K (GSM8K) is used as the attribution target. For ease of comparison, we denote the default in-domain setting as **In.D** and the cross-domain setting as **Crs.D**. Since most other attribution methods fail to operate reliably under the cross-domain scenario, we report only the results of different combinations within GTA. Other dataset configuration and training/attribution pipeline is the same as in the paper. The results are given below:
>
> |||GSM8K||||CodeAplaca-20K||||
> |-|-|-|-|-|-|-|-|-|-|
> |Information|Setting|AUC|TPR@5%FPR|TPR@1%FPR|Acc.|AUC|TPR@5%FPR|TPR@1%FPR|Acc.|
> |confidence|**In.D**|**64.50**|**13.60**|**3.70**|**60.79**|**53.86**|**5.83**|**1.04**|**53.19**|
> ||Crs.D|59.27|9.20|1.60|57.29|52.84|3.40|0.60|52.17|
> |filtered confidence|**In.D**|**59.53**|**8.83**|**1.87**|**58.14**|**54.13**|**6.03**|**1.02**|**53.36**|
> ||Crs.D|56.90|6.68|1.53|54.45|52.78|4.46|1.25|52.60|
> |**DDM** (ours)|**In.D**|**66.91**|**14.50**|**4.68**|**64.92**|**62.64**|**6.38**|**1.31**|**61.78**|
> ||Crs.D|62.38|12.78|3.12|60.16|58.32|5.97|1.13|57.39|
>
> As shown in the results, the cross-domain setting is clearly more challenging. While performance does decrease, the drop remains modest, typically within 5%. This indicates that DDM-GTA exhibits strong resilience under cross-domain conditions.
>
> - For adversarial imitation of DDM patterns, we consider degrading the structural information in the DDM by **injecting Gaussian noise**. Specifically, we introduce **two noise levels** where σ=0.3 and σ=0.5, respectively. All other dataset configurations and the training/attribution pipeline remain identical to those described in the main paper. We set the token length to 32 and block size to 16. The results on LLaDA are shown below:
>
> ||GSM8K||||CodeAplaca-20K||||
> |-|-|-|-|-|-|-|-|-|
> |Noise Level|AUC|TPR@5%FPR|TPR@1%FPR|Acc.|AUC|TPR@5%FPR|TPR@1%FPR|Acc.|
> |σ=0.0|**66.91**|**14.50**|**4.68**|**64.92**|**62.64**|**6.38**|**1.31**|**61.78**|
> |σ=0.3|65.81|8.83|1.92|62.69|60.78|5.69|1.22|59.36|
> |σ=0.5|63.77|7.28|1.80|60.50|57.92|4.48|1.18|57.52|
>
> We observe that the attribution performance degrades to some extent when the constructed DDMs are adversarially perturbed with Gaussian noise. However, even under relatively **large noise levels** (σ=0.5), the performance **remains reasonably strong** (AUC remains around 60%).
>
> ---
>
> Thanks again for the thoughtful and constructive feedback. We would definitely love to further interact with the reviewer if there are any further questions.

---

> ### Author Response · Authors · 2025-11-27
>
> Dear Reviewer gzXx,
>
> Thanks again for the valuable comments and encouraging feedback. As the discussion period is coming to a close soon, we would very appreciate it if you could let us know if our response has sufficiently addressed your concerns, or if there are any remaining questions we can clarify.
>
> We also revised the manuscript based on your comments. If there is anything you are not satisfied with, please let us know, we will be more than happy to revise the paper accordingly.

---

> > ### Comment · Reviewer_gzXx · 2025-11-28
> >
> > Thank you for the clarifications. I have carefully read your discussion and incorporated your points into my updated assessment.

---

### Official Review · Reviewer_UBPg · 2025-11-08

**Soundness:** 3
**Presentation:** 2
**Contribution:** 2
**Rating:** 4
**Confidence:** 4

**Summary:**

This paper investigates the problem of model attribution, with a focus on diffusion language models as an emerging architecture. It proposes the design of DMM along with a corresponding GTA approach for second-order modeling and learning, which is validated and analyzed on the Dream and LLADA models.

**Strengths:**

- This work represents one of the first efforts to address model attribution from the perspective of diffusion models and demonstrates the effectiveness of the proposed approach.
- The methodology is straightforward and built upon a reasonable foundation.

**Weaknesses:**

- As the study focuses on second-order modeling and analysis of different decoding strategies, it lacks exploration of recent blockwise diffusion models (e.g., SDAR[1], dllm-var[2]). The applicability of the proposed method to such models remains unclear.
- Beyond mathematical and code reasoning tasks, the evaluation does not assess performance on knowledge-intensive tasks, leaving the method’s effectiveness in such domains unverified.
- The generalization capability of the method, along with the impact of key hyperparameters, requires further analysis and insight.

[1] SDAR: A Synergistic Diffusion-AutoRegression Paradigm for Scalable Sequence Generation

[2] Diffusion LLM with Native Variable Generation Lengths: Let [EOS] Lead the Way

**Questions:**

- Given that the dominant model architecture remains autoregressive, is the proposed method suitable for autoregressive decoding schemes? How can it be efficiently adapted to autoregressive models?

---

> ### Author Response · Authors · 2025-11-20
> **Author Response [1/3]**
>
> It’s our great honor to receive Reviewer UBPg’s valuable comments and kind words to our work. We would like to address all the concerns as below.
>
> ---
>
> >**[W1]:** Explore the applicability of DDM and GTA on blockwise diffusion models (e.g., SDAR[1], dllm-var[2]).
>
> Thanks for the valuable comment. We further evaluate the performance of DDM and GTA using the two mentioned models: **SDAR-30B-A3B-Chat** [1] and **dLLM-Var** [2]. The dataset configuration and training/attribution pipeline is the same as in the paper. We set the token length and block size to 32 and 16, respectively. For SDAR-30B-A3B-Chat, we adopt the static decoding strategy. For dLLM-Var, we use the default decoding setup. Results under the **most challenging CCA setting** are given below:
>
> - **SDAR-30B-A3B-Chat** [1]:
>
> ||||GSM8K||||CodeAlpaca-20K||||
> |-|-|-|-|-|-|-|-|-|-|-|
> |Scenario|Method|Information|AUC|TPR@5%FPR|TPR@1%FPR|Acc.|AUC|TPR@5%FPR|TPR@1%FPR|Acc.|
> |CCA|Perplexity|Perplexity|51.23|5.27|1.35|52.79|49.93|4.54|1.03|51.46|
> ||Clustering|confidence|52.31|5.42|1.48|52.79|52.38|5.21|2.59|52.51|
> |||filtered confidence|52.15|6.83|1.92|53.03|51.97|4.89|2.46|52.49|
> |||**DDM**|**55.27**|**8.27**|**2.68**|**57.56**|**54.78**|**7.98**|**3.12**|**54.06**|
> ||Distance|confidence|58.86|4.76|1.37|56.54|57.78|6.52|2.40|57.51|
> ||| filtered confidence|57.69|3.52|1.62|54.91|56.83|4.93|2.84|56.82|
> ||| **DDM**|**63.81**|**9.31**|**2.33**|**60.83**|**60.28**|**12.36**|**5.68**|**58.84**|
> ||**GTA**|confidence|61.59|15.72|3.65|60.46|55.41|6.73|3.64|54.08|
> |||filtered confidence|59.13|13.52|2.86|59.31|54.52|7.32|5.34|54.04|
> |||**DDM** (ours)|**65.98**|**17.65**|**4.38**|**64.24**|**62.87**|**12.59**|**7.03**|**61.68**|
>
> - **dLLM-Var** [2]:
>
> ||||GSM8K||||CodeAlpaca-20K||||
> |-|-|-|-|-|-|-|-|-|-|-|
> |Scenario|Method|Information|AUC|TPR@5%FPR|TPR@1%FPR|Acc.|AUC|TPR@5%FPR|TPR@1%FPR|Acc.|
> |CCA|Perplexity|Perplexity|51.47|3.68|1.67|51.65|48.65|4.65|1.31|51.06|
> ||Clustering|confidence|51.42|4.46|0.92|50.97|51.65|4.28|1.09|51.84|
> |||filtered confidence|52.54|6.62|1.32|52.08|51.20|3.86|1.68|51.49|
> |||**DDM**|**56.89**|**7.68**|**1.95**|**56.06**|**55.39**|**5.63**|**1.64**|**54.40**|
> ||Distance|confidence|57.45|3.85|1.06|56.37|58.62|4.58|1.45|58.25|
> ||| filtered confidence|59.65|3.40|0.96|58.92|58.65|3.53|1.34|58.04|
> ||| **DDM**|**62.96**|**8.32**|**2.73**|**61.65**|**61.75**|**10.98**|**4.95**|**60.13**|
> ||**GTA**|confidence|60.73|11.48|2.86|60.32|59.41|5.86|2.96|59.10|
> |||filtered confidence|61.66|12.68|4.05|60.08|60.56|6.30|2.58|59.36|
> |||**DDM** (ours)|**67.06**|**15.84**|**6.26**|**65.28**|**65.25**|**11.06**|**5.38**|**63.50**|
>
> As can be observed, DDM together with GTA **remains the most useful** attribution method on these two models. Also, when DDM is cooperated with other attribution methods, it can still help to improve the performance.  Besides, we can observe that for SDAR-30B-A3B-Chat, the attribution performance is particularly strong in the **low-FPR regime (TPR@Low FPR)**. For example, the GPA-DDM combination achieves over 7% TPR@1% FPR on the CodeAlpaca-20K dataset. This indicates that the model retains **highly discriminative attribution signals** even under extremely low false-positive constraints.
>
> We sincerely hope that our response can alleviate this concern and we will add relative discussions on these two models in the revision.
>
> ---
>
> [1] SDAR: A Synergistic Diffusion-AutoRegression Paradigm for Scalable Sequence Generation
>
> [2] Diffusion LLM with Native Variable Generation Lengths: Let [EOS] Lead the Way

---

> ### Author Response · Authors · 2025-11-20
> **Author Response [2/3]**
>
> > **[W2]:** Beyond mathematical and code reasoning tasks, evaluate the performance on knowledge-intensive tasks.
>
> Thanks for the valuable suggestion. We further evaluate the performance on **GPQA** [3], which is a popular knowledge-intensive benchmark. We set the token length and block size to 64 and 16, respectively. We use the gpqa_main subset for DDM construction and the gpqa_diamond subset is used as the attribution target. The results are given below:
>
> |||| LLaDA ||||Dream||||
> |-|-|-|-|-|-|-|-|-|-|-|
> |Scenario|Method|Information|AUC|TPR@5%FPR|TPR@1%FPR|Acc.|AUC|TPR@5%FPR|TPR@1%FPR|Acc.|
> |CCA|Perplexity|Perplexity|53.65|5.68|2.06|52.65|51.88|4.95|1.38|51.62|
> ||Clustering|confidence|54.45|6.42|2.86|53.02|53.66|4.28|2.05|53.08|
> |||filtered confidence|53.08|5.02|1.37|51.68|52.68|3.72|1.27|52.45|
> |||**DDM**|**55.48**|**6.79**|**3.02**|**54.38**|**54.72**|**6.28**|**3.16**|**55.10**|
> ||Distance|confidence|58.42|4.62|1.62|56.77|56.20|5.83|1.95|54.89|
> |||filtered confidence|59.02|5.98|2.64|57.90|57.59|4.57|2.32|57.30|
> |||**DDM**|**61.24**|**6.12**|**3.08**|**59.68**|**59.24**|**8.35**|**4.14**|**58.90**|
> ||**GTA**|confidence|63.38|9.69|4.15|62.14|59.02|6.37|3.69|57.88|
> |||filtered confidence|62.69|8.96|3.98|61.02|57.93|6.04|2.72|57.10|
> |||**DDM** (ours)|**65.83**|**11.85**|**5.62**|**64.30**|**63.32**|**10.82**|**4.98**|**62.70**|
>
> From the results, we can observe that for Dream (initialized from an AR-based model and then finetuned in a discrete diffusion way), it is more difficult to attribute on tasks like GPQA compared to LLaDA. A plausible explanation is that, for such **complex reasoning tasks**, Dream tends to exhibit stronger AR-like behavior, i.e., decoding from left to right, and therefore providing weaker bi-directional structural information for attribution. Nevertheless, in terms of overall performance, the GTA-DDM combination still achieves the strongest attribution results, and DDM continues to enhance the performance of other attribution methods as well.
>
> ---
>
> [3] Gpqa: A graduate-level google-proof q&a benchmark

---

> ### Author Response · Authors · 2025-11-20
> **Author Response [3/3]**
>
> > **[W3]:**  Further analysis on the generalization capability and the impact of key hyperparameters.
>
> Thanks for highlighting this important consideration. We conduct two experiments in respond to this concern:
>
> - Regarding the **generalization capability,** we evaluate the cross-domain robustness of our method, i.e., how well it performs when the target dataset differs from the one used for model training. In this setting, we use GSM8K (CodeAlpaca-20K) for model training and CodeAlpaca-20K (GSM8K) is used as the attribution target. For ease of comparison, we denote the default in-domain setting as **In.D** and the cross-domain setting as **Crs.D**. Since most other attribution methods fail to operate reliably under the cross-domain scenario, we report only the results of different combinations within GTA. Other dataset configuration and training/attribution pipeline is the same as in the paper. The results are given below:
>
> |||GSM8K||||CodeAplaca-20K||||
> |-|-|-|-|-|-|-|-|-|-|
> |Information|Setting|AUC|TPR@5%FPR|TPR@1%FPR|Acc.|AUC|TPR@5%FPR|TPR@1%FPR|Acc.|
> |confidence|**In.D**|**64.50**|**13.60**|**3.70**|**60.79**|**53.86**|**5.83**|**1.04**|**53.19**|
> ||Crs.D|59.27|9.20|1.60|57.29|52.84|3.40|0.60|52.17|
> |filtered confidence|**In.D**|**59.53**|**8.83**|**1.87**|**58.14**|**54.13**|**6.03**|**1.02**|**53.36**|
> ||Crs.D|56.90|6.68|1.53|54.45|52.78|4.46|1.25|52.60|
> |**DDM** (ours)|**In.D**|**66.91**|**14.50**|**4.68**|**64.92**|**62.64**|**6.38**|**1.31**|**61.78**|
> ||Crs.D|62.38|12.78|3.12|60.16|58.32|5.97|1.13|57.39|
>
> As shown in the results, the cross-domain setting is clearly more challenging. While performance does decrease, the drop remains modest, typically within 5%. This indicates that DDM-GTA exhibits strong resilience under cross-domain conditions.
>
> - Regarding **hyperparameters**, we conduct a deeper investigation into the **effect values**. To eliminate **scale-related biases**, we sample three distinct values from a same range level to serve as the effect values, and three different range levels are used: (0,10.0], (0,100.0], and (0,1000.0]. For each range level, we perform five random samplings and report the averaged results. We set the token length and block size to 32 and 16, respectively. The results under GSM8K and CodeAlpaca-20K are summarized below:
>
> |_GSM8K_|Default|(0, 10.0]|(0, 100.0]|(0, 1000.0]|_CodeAlpaca-20K_|Default|(0, 10.0]|(0, 100.0]|(0, 1000.0]|
> |-|-|-|-|-|-|-|-|-|-|
> |CMA|99.95|99.87±0.06|99.79±0.10|99.65±0.19|CMA|98.94|98.84±0.08|98.76±0.13|98.62±0.18|
> |IRA|81.75|81.84±0.16|81.59±0.19|81.37±0.26|IRA|65.05|64.96±0.12|64.79±0.18|64.63±0.23|
> |CCA|66.91|66.59±0.24|66.44±0.27|66.40±0.30|CCA|62.64|62.55±0.19|62.47±0.26|62.32±0.29|
>
> It can be observed that the choice of the effect value **has only a negligible impact on performance**. Conceptually, the effect value simply serves as a **positional marker for different signal types**. Nevertheless, some patterns can still be observed from the results. For example, random sampling of the effect value remains effective, but restricting the sampling range to **a smaller interval** leads to more stable outcomes and yields slight (though marginal) performance improvements. We will include the corresponding results and discussion in the revision.
>
> ---
>
> > **[Q1]:** Given that the dominant model architecture remains autoregressive, is the proposed method suitable for autoregressive decoding schemes? How can it be efficiently adapted to autoregressive models?
>
> Thanks for the insightful feedback. Our method is intentionally designed for dLLMs, leveraging their unique **bi-directional decoding** process. That said, exploring its transferability to AR-based LLMs is indeed valuable, and there may be several key challenges.
>
> For example, AR-based LLMs do not have a bi-directional decoding mechanism. Their **left-to-right** decoding trajectory does not naturally encode structured information, nor does it provide inter-step dependencies across future and past positions. Consequently, the only usable signal comes from the final full generated sequence. A straightforward adaptation would be to apply GTA directly to the model’s **final confidence distribution**, comparing different models’ reactions to the same prompt. Going further, since AR-based LLMs are trained with a **maximum-likelihood** objective, their response confidence over a prompt may exhibit local maxima and other positional patterns. Capturing such localized features before applying GTA for estimation could lead to better results.
>
> Overall, we truly appreciate the reviewer’s suggestion on exploring a unified attribution method applicable to both AR-based models and dLLMs. We see this as a valuable and exciting direction and plan to pursue it further in future work.
>
>
>
> ---
>
> Thanks again for the thoughtful and constructive feedback. If reviewer UBPg has any remaining concerns, we would definitely love to clarify.

---

> > ### Author Response · Authors · 2025-11-27
> >
> > Dear Reviewer UBPg,
> >
> > Thanks again for your valuable comments and kind words to our work. If there is any further questions, or if there is anything we have not fully addressed, please let us know. We would be more than happy to clarify or improve anything that may still be unclear.
> >
> > We also include discussions on the two works you kindly pointed out (SDAR and dLLM-Var) in our revision (line 107). If there is anything you are not satisfied with, we would be very happy to make further improvements.

---

### Author Response · Authors · 2025-11-20

We sincerely thank all the reviewers for their time and insightful feedback. We appreciate reviewers for their kind words regarding the novelty and utility of our work: "straightforward and reasonable method" (Reviewer **UBPg**); "clear, novel framing, simple, lightweight, and gray-/black-box compatible" (Reviewer **gzXx**); "experimentally comprehensive" (Reviewer **dGL7**); "interesting and timely research direction" (Reviewer **3QFe**).

Most of the reviewers’ concerns focus on _further exploring the scalability and effectiveness of our method_ (for example, the efficiency of our method (**dGL7**, **3QFe**), results on more models and datasets (**UBPg**, **3QFe**), performance under different hyperparameters and settings (**UBPg**, **gzXx**, **dGL7**, **3QFe**)), as well as _clarification of several points that were not sufficiently explained_ in the original submission (**gzXx**, **dGL7**, **3QFe**). We give our response to these concerns point by point in our Author Response to each reviewer, and **we have incorporated the necessary changes into the revised manuscript**. All modifications are highlighted in blue for ease of reference.

We sincerely hope that our responses could address the reviewers’ concerns. **If the reviewers have any remaining specific comments or suggestions, we would be very happy to provide further clarification and response.**

---

### Meta-Review · Area_Chair_Ynaq · 2025-12-10

**Summary:**

The major concerns that led to the final decision are that the method requires access to the confidence scores of the entire decoding process and is tailored for diffusion LLMs. In addition, the experiments are restricted to two small models and lack results in several settings.

**Reviewer Concerns:**

The lack of experiments in several settings is mostly addressed by the rebuttal. However, the major concern that the method requires access to the entire decoding process and is restricted for diffusion LLMs is not fully addressed. Even though additional experiments of gray-box setting were provided by the rebuttal, but it is not clear what it means by “GTA constructs distributions directly from the model’s decoding history” if only the model outputs are available. It’s not clear if it generalizes to gray-/black-box settings and the performance is not good either.

**Reviewer Scores:**

Reviewer dGL7 and Reviewer gzXx may increase the final scores a bit since their main concerns were about additional experiments and some clarification of the setups. Reviewer UBPg and Reviewer 3QFe may not increase the scores as the question of generalizability and white-box setting were not addressed.

---

### Decision · Program_Chairs · 2026-01-26

Reject